# Climate change and land use threaten global hotspots of phylogenetic endemism for trees

Wen-Yong Guo ®[1,2,3] ✉, Josep M. Serra-Diaz[4,5], Wolf L. Eiserhardt ®[3],
Brian S. Maitner ®[6], Cory Merow ®[4], Cyrille Violle ®[7], Matthew J. Pound ®[8],
Miao Sun[3,9], Ferry Slik[10], Anne Blach-Overgaard ®[2,3], Brian J. Enquist[6,11] &
Jens-Christian Svenning ®[2,3]

Across the globe, tree species are under high anthropogenic pressure. Risks of extinction are notably more severe for species with restricted ranges and distinct evolutionary histories. Here, we use a global dataset covering 41,835 species (65.1% of known tree species) to assess the spatial pattern of tree species' phylogenetic endemism, its macroecological drivers, and how future pressures may affect the conservation status of the identified hotspots. We found that low-to-mid latitudes host most endemism hotspots, with current climate being the strongest driver, and climatic stability across thousands to millions of years back in time as a major co-determinant. These hotspots are mostly located outside of protected areas and face relatively high land-use change and future climate change pressure. Our study highlights the risk from climate change for tree diversity and the necessity to strengthen conservation and restoration actions in global hotspots of phylogenetic endemism for trees to avoid major future losses of tree diversity.

Trees are pivotal to the biosphere and human well-being, e.g., via carbon sequestration and habitat provision for plants and animals[1–5]. However, globally, the majority of tree species are under pressure from anthropogenic activities[6,7], notably habitat conversion and loss, overexploitation, and biological invasions[8–12]. Although about half of the average tree species' ranges are protected in existing protected areas, only about a quarter of the range-restricted tree species' ranges are protected on average, among which, more than 6000 species' ranges are entirely outside of existing protected areas[7]. Due to their unique ecological and evolutionary characteristics[13], range-restricted

species, or endemics, are often used to guide conservation prioritization because of their inherent high risk of extinction[14–16].

Geographically rare and evolutionarily irreplaceable lineages have a relatively high likelihood of possessing unique functional attributes, thus are crucial for multidimensional (i.e., taxonomic, functional and phylogenetic) biodiversity, ecosystem function and services[13,17]. To capture the evolutionary rarity within a given area, it has recently been proposed to quantify the degree to which phylogenetic diversity[18,19] is restricted to a particular geographic area, i.e., phylogenetic endemism (PE)[13,20,21]. In addition to not always exhibiting similar geographic

[1]Research Center for Global Change and Complex Ecosystems & Zhejiang Tiantong Forest Ecosystem National Observation and Research Station, School of Ecological and Environmental Sciences, East China Normal University, 200241 Shanghai, P. R. China. [2]Center for Ecological Dynamics in a Novel Biosphere (ECONOVO) & Center for Biodiversity Dynamics in a Changing World (BIOCHANGE), Department of Biology, Aarhus University, 8000 Aarhus C, Denmark. [3]Section for Ecoinformatics & Biodiversity, Department of Biology, Aarhus University, 8000 Aarhus C, Denmark. [4]Eversource Energy Center and Department of Ecology and Evolutionary Biology, University of Connecticut, Storrs, CT, USA. [5]Université de Lorraine, AgroParisTech, INRAE, Silva, Nancy, France. [6]Department of Ecology and Evolutionary Biology, University of Arizona, Tucson, AZ 85721, USA. [7]CEFE, Univ Montpellier, CNRS, EPHE, IRD, Montpellier, France. [8]Department of Geography and Environmental Sciences, Northumbria University, Newcastle upon Tyne NE1 8ST, United Kingdom. [9]National Key Laboratory for Germplasm Innovation & Utilization of Horticultural Crops, Huazhong Agricultural University, Wuhan 430070, P. R. China. [10]Environmental and Life Sciences, Faculty of Science, Universiti Brunei Darussalam, Jalan Tungku Link, BE1410 Gadong, Brunei Darussalam. [11]The Santa Fe Institute, 1399 Hyde Park Rd, Santa Fe, NM 87501, USA. ✉e-mail: guowyhgy@gmail.com

patterns to species endemism[22–25], by capturing additional evolutionary information of the biodiversity patterns, PE can be used to distinguish whether taxa are ancient or recently diverged, i.e., paleo- or neo-endemism, based on an approach called categorical analysis of neo- and paleo-endemism (CANAPE)[23,25]. Specifically, areas of paleo-endemism represent potential biodiversity centers that harbour lineages, which diverged or immigrated relatively deep into the past, but became extinct elsewhere while persisting within these regions. In contrast, neo-endemism characterizes biodiversity centers where recently diverged lineages are concentrated[16,22,25]. For instance, Mesoamerica and the South American Atlantic Forest act as neo-endemism or paleo-endemism centers for butterflies, respectively[26], while for leaf beetles, tropical forests in general are centers for both paleo- and neo-endemism, i.e., mixed endemism centers[27]. Overall, areas with high PE, whether paleo- or neo-endemism centers, are important to consider in conservation planning[23,28]. Paleo- and neo-endemism centers have been studied for well-examined major organism groups such as vertebrates[23], as well as narrower taxonomic groups of certain special interest within trees, e.g., Australian eucalypts[29] and acacias[25]. However, so far, the global occurrence of paleo- and neo-endemism centers for tree species overall remain poorly known[30].

Previous studies have emphasized the importance of current climate conditions as critical determinants of species diversity and endemism (e.g.,[31–33]). However, current patterns of biodiversity may also reflect legacies of past climates that influenced speciation, extinction, and dispersal[34–40], and thereby left imprints on PE[37,38]. Over geological time scales, Earth's climate has undergone dramatical changes[41], such as cooling or warming trends and episodes, including major climatic transitions that have coincided with global ecosystem shifts[42–46]. Regions characterized by long-term climatic stability over geological time scales have played a substantial role in driving high rates of speciation and low rates of extinction[42,44]. These conditions, in turn, increase the likelihood of immigration, resulting in higher levels of PE and the presence of either paleo-endemism or neo-endemism centers[23,24,30,37,47,48]. Conversely, regions that have experienced pronounced climate oscillations, such as those occurring during glacial-interglacial cycles, are expected to exhibit both high speciation and extinction rates[49]. This dynamic leads to substantial species turnover, and reduced chance for immigration, thereby could contribute to the formation of neo-endemism centers[34,37,38,47,49,50]. Up to now, an explicit test of these hypotheses for global tree PE centers is missing, limiting our understanding of the vulnerability of tree PE hotspots, areas of long-term importance for tree survival and diversification, to current human[7,51] and future climate change threats[52,53], and our ability to develop efficient conservation planning.

Here, we use a recently compiled global tree distribution dataset[54], covering 41,835 species or 65.1% of the known tree species globally[55], to map tree PE and explore its macroecological drivers, its conservation status, and exposure to future pressures. We analysed the relative roles of present climate conditions and past climate variability at various time scales in shaping global tree PE patterns. To assess the potential effects of paleoclimatic change on PE, we examined the relative importance of two paleoclimatic time frames while accounting for a broad range of likely contemporary drivers, such as current climate and topography (Table S1). Specifically, we explored the influence of paleoclimate of the late Cenozoic – the time period where current species diversity in large part originated[34,36–39] via climate estimates for two epochs: the warm and humid late Miocene, ca. 11.63 – 7.25 Mya; and the cold and dry Pleistocene glaciations (represented by the Last Glacial Maximum, LGM - 21 kya) (Table S2, Figs. S1 and S2). A link between hotspots of PE (i.e., areas with high PE) and long-term climate variability would support that tree diversity can be expected to be sensitive to future climate change, if the stability in these hotspots is not maintained. In addition, we investigated how

current human activity intensity and future climate change (Figs. S1 and S2) impact tree PE hotspots, where human activity intensity was represented via the Human Modification Index, a cumulative spatial layer integrating 13 different types of human activities, such as human population density, pathways, and croplands[51]. Finally, we determined how well tree PE hotspots are covered by the existing protected areas as well as three tree-specific diversity conservation prioritization frameworks[7], i.e., the top 17%, top 30%, and top 50% tree diversity priority areas, which were developed by prioritizing multiple-dimensional (i.e., taxonomic, phylogenetic, and functional dimensions) tree diversity globally[7]. Each of the top 17%, top 30%, and top 50% priority areas corresponds to the Convention on Biological Diversity (CBD) 2020 protected areas target, the COP15 30×30 target agreed by 188 governments on 19 December 2022, aiming to conserve at least 30% of Earth's surface by 2030, and the Half-Earth target[56], a global goal in the 2050 Vision for Biodiversity agreed at COP15.

In this work, we reveal that regions with high phylogenetic endemism among tree species are primarily located at low-to-mid latitudes, and these endemism hotspots are not only linked to current environmental conditions, but also associated with regions characterized by long-term climate stability. We show that these hotspots currently receive insufficient protection from existing protected areas and are already under significant human pressure, while also facing substantial risks from impending climate change. These results highlight the urgency of an ambitious expansion of global protected areas to mitigate these rising risks and enhance overall conservation efforts for safeguarding the diversity of tree species worldwide.

## Results
### Global patterns of tree phylogenetic endemism and rarity hotspots

Angiosperm trees (broadleaved trees) and gymnosperm trees (mostly conifers) exhibit distinct spatial PE patterns, besides New Guinea and New Caledonia, where PE is high for both groups (Fig. 1 and S3). High angiosperm PE mainly occurs in Central America, northern Andes, the east coast of Madagascar, southwestern China, and northern Borneo and Peninsula Malaysia (Fig. 1a). In contrast, high gymnosperm PE areas occur primarily in southern China, northeastern Borneo, Japan, Tasmania, and Fiji (Fig. 1b).

About 24.3% of angiosperm (1508 of 6198) and 12.0% of gymnosperm (145 of 1200) distribution grid cells (at a 110 ×110 km resolution) were identified as PE hotspots, i.e., centers of neo-, paleo-, and mixed PE (Fig. 2). Within the 24.3% PE hotspots for angiosperm trees, 22.9% (1419 cells) are mixed centers with both neo- and paleo-endemics, distributed in the Andean region of South America, Africa, South Asia, and Australia. Angiosperm centers for only neo- or only paleo-endemism are relatively rare−0.8% (49 cells) and 0.7% (40 cells), respectively, and mostly scattered around the mixed-endemism centers (Fig. 2a). The same is true for gymnosperms. For gymnosperms, centers of only paleo-endemism account for just 0.7% (8 cells) of gymnosperm global distribution, mainly occurring in southeastern China. Sites dominated by neo-endemism, accounting for 1.2% (14 cells) of the gymnosperm global distribution, are scattered in the mountainous areas of southwestern China (Fig. 2b). Mixed-endemism gymnosperm centers are more frequent (10.3%, 123 cells) and occur predominantly in southern China, Japan, northern Borneo, Papua New Guinea, the east coast of Australia, and New Zealand. In total, 128 grid cells of PE hotspots are shared by angiosperm and gymnosperm species, consisting each of 8.5% and 88.3% of angiosperm and gymnosperm endemism hotspots (Fig. 2c, d), i.e., gymnosperm endemism hotspots are mostly also angiosperm PE hotspots, but not vice versa. The shared hotspots are located in Southeast Asia, southwest mountainous areas of China, northern Borneo, Papua New Guinea, the east coast of Australia, and New Zealand (Fig. 2c). Globally, 20.8% of grid

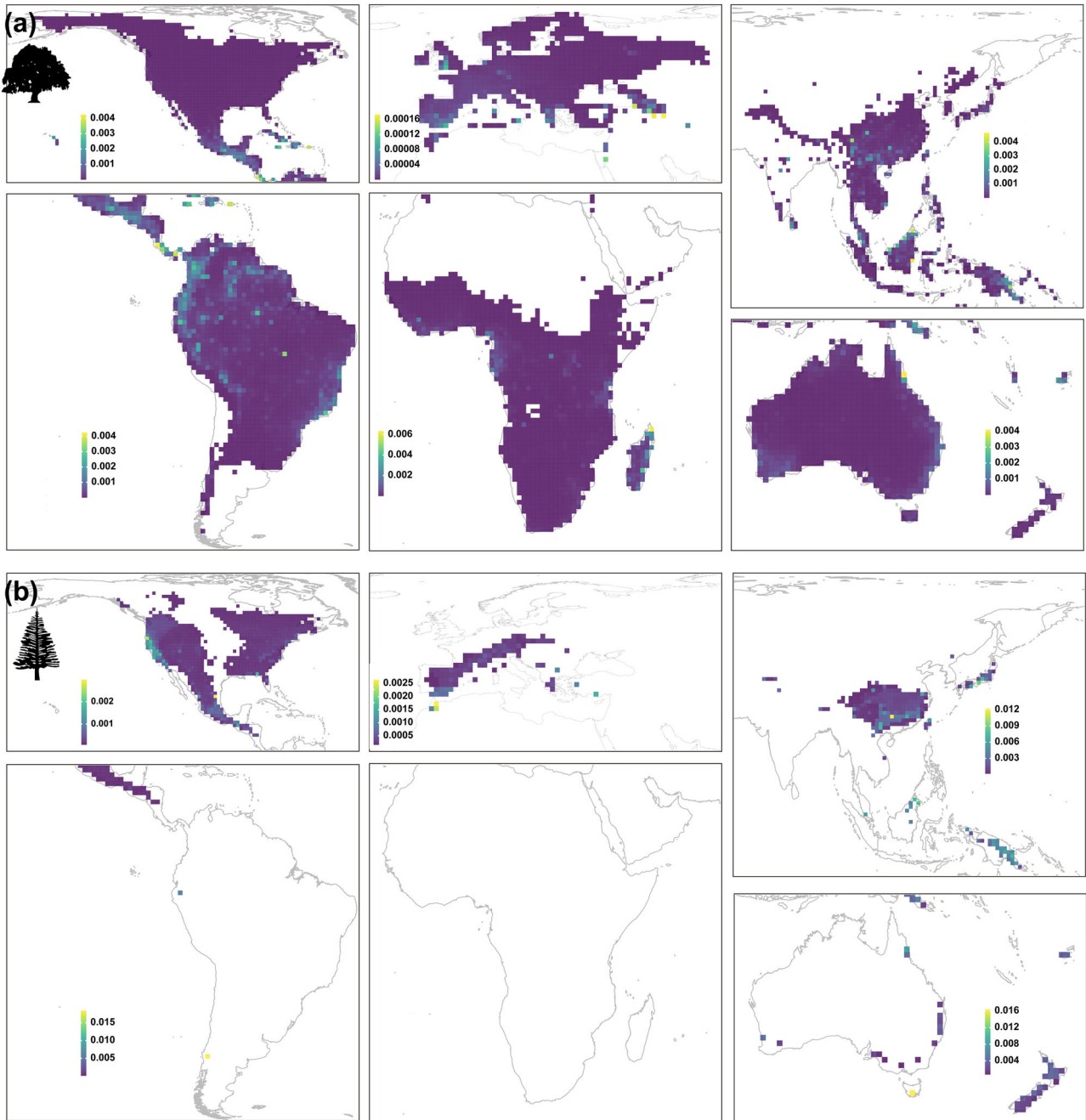

**Fig. 1 | Observed phylogenetic endemism for global tree species.** (**a**) angiosperm and (**b**) gymnosperm trees. Pictograms courtesy of PhyloPic (www.phylopic.org): (**a**) Tracy A. Heath; (**b**) T. Michael Keesey.

cells (with more than five tree species) were detected as PE hotspots for angiosperms, gymnosperms, or both (Fig. 2c, d).

### Drivers of global tree phylogenetic endemism and rarity hotspots

Accounting for spatial autocorrelation, simultaneous autoregressive models (SARs) explained more than 90% (angiosperm) or 84% (gymnosperm) of the variation in PE (Tables S3 and S4). Present-day annual precipitation (AP) emerged as a dominant factor, with either the strongest or the second strongest standardized effect, with a positive relation to PE for both gymnosperms and angiosperms (Fig. 3a, b). However, present-day mean annual temperature (MAT) had an even stronger, positive association with PE in the case of angiosperms. Furthermore, the elevation range also has consistent positive

associations with PE for both angiosperms and gymnosperms. With respect to the paleoclimatic variables, LGM AP and MAT anomalies (i.e., LGM AP/MAT minus present AP/MAT) showed consistent positive relations to PE for both groups, indicating that high PE is associated to relatively warm and wet LGM conditions. Both Miocene anomalies show no relation to gymnosperm PE. On the other hand, Miocene MAT and AP anomaly exhibited positive and negative associations with angiosperm PE, respectively, indicating warmer regions with less precipitation during the Miocene than at present have higher angiosperm PE.

Comparing the combined endemism hotspots (i.e., neo-, paleo-, and mixed endemism centers, as shown in Fig. 2a, b) to non-hotspots (not significant regions in Fig. 2a, b), both angiosperm (Fig. 3c) and gymnosperm (Fig. 3d) hotspots exhibited greater elevation ranges,

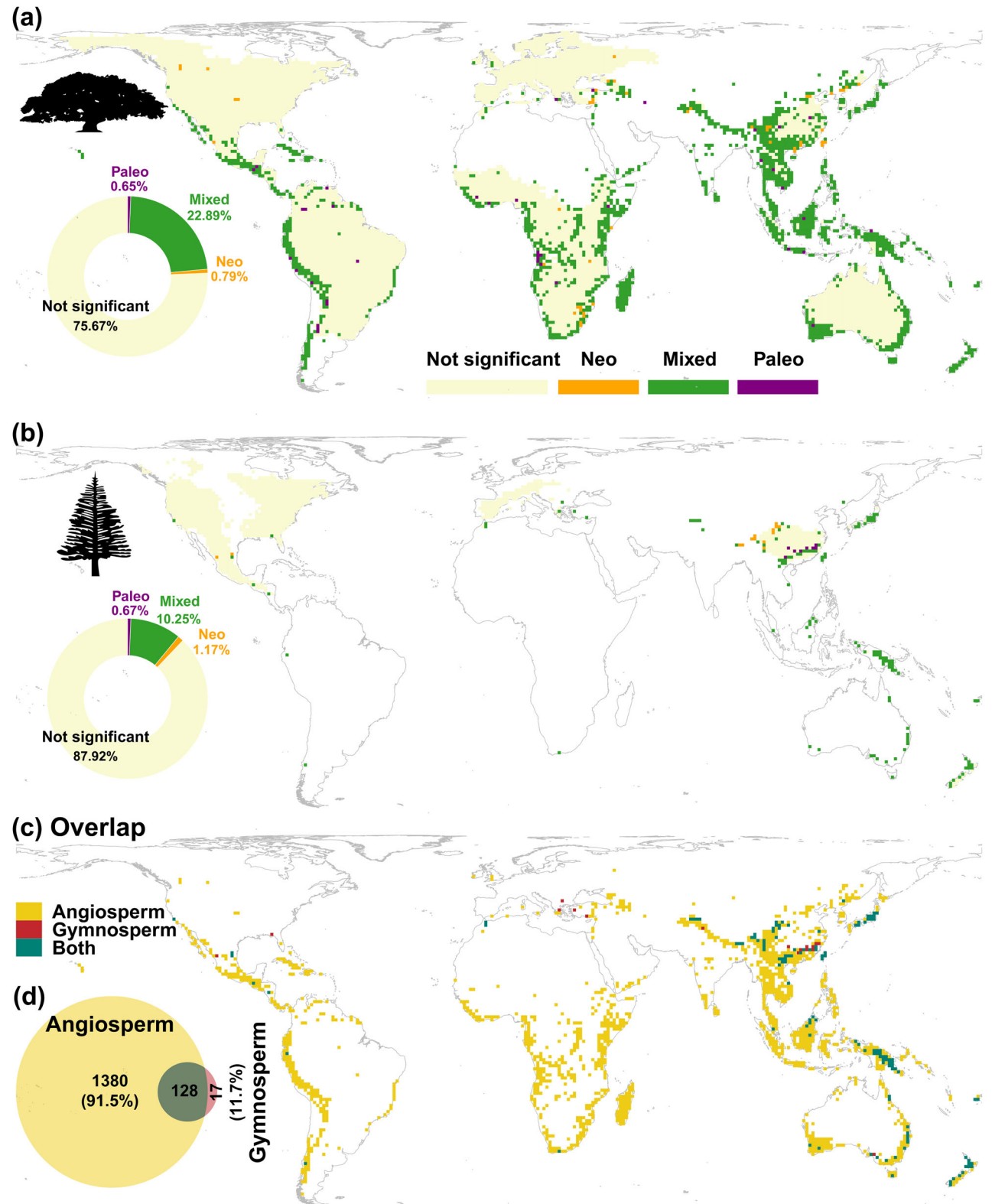

**Fig. 2 | Global distribution of tree endemism types.** (**a**) angiosperm, (**b**) gymnosperm trees, and (**c**) their overlap. Centers of neo-endemism (Neo, i.e., concentrations of rare short branches), paleo-endemism (Paleo, i.e., concentrations of rare long branches), and mixed neo- and paleo-endemism (Mixed) were identified using a randomization analysis [categorical analysis of neo- and paleo-endemism (CANAPE)] conducted separately for angiosperm and gymnosperm trees. (**d**) Venn diagram showing the area and percentage overlap of the significant endemism regions (i.e., hotspot, including centers of either neo-endemism, paleo-endemism, or mixed neo- and paleo-endemism) between angiosperm and gymnosperm trees. Numbers are sum of the hotspot cells, and the percentages in brackets correspond to the percentages of non-overlapping cells in each of the angiosperm and gymnosperm hotspots. Pictograms courtesy of PhyloPic (www.phylopic.org): (**a**) Tracy A. Heath; (**b**) T. Michael Keesey.

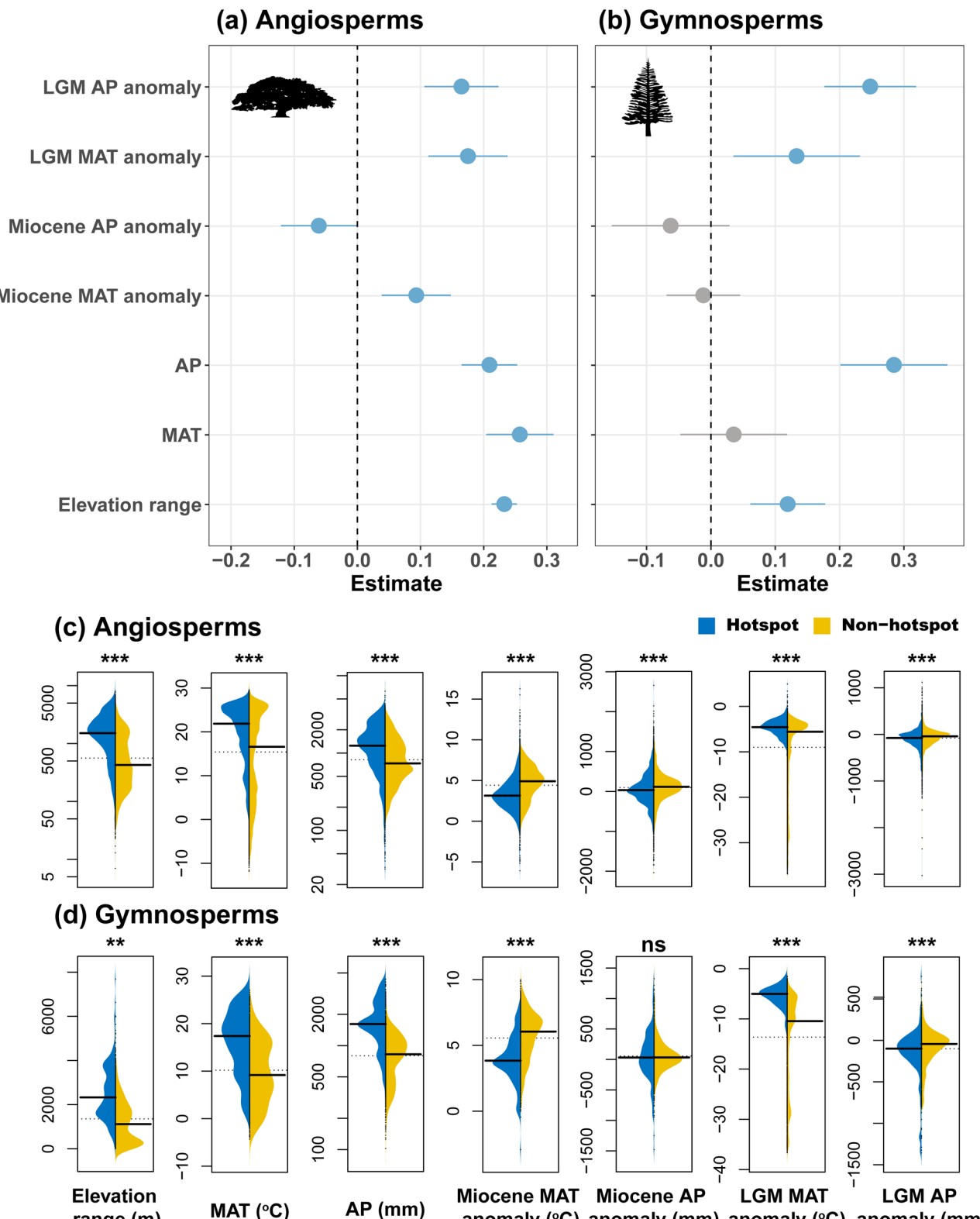

**Fig. 3 | Determinants of global tree phylogenetic endemism.** (**a**) angiosperm and (**b**) gymnosperm trees. Error bars represent estimates (standardized slopes) and 95% confidence intervals (C.I.s) which were obtained from the best spatial auto-regressive models. Grey indicates non-significant estimates, i.e., their 95% C.I.s overlapping with zero. Results from non-spatial linear regression analyses are presented in Fig. S4. (**c**) and (**d**) represent significance tests between endemism hotspots (i.e., combined neo-, paleo-, and mixed endemism centers in Fig. 2a, b, $n$ = 1536 & 151 for angiosperms and gymnosperms, respectively) and non-hotspots (not significant regions in Fig. 2a, b, $n$ = 5820 & 1341 for angiosperms and gymnosperms, respectively) for each of the environmental variables. Significance tests were carried out using the two-sample Fisher-Pitman permutation test (100,000 permutations, *** represents $p < 0.001$; ** represents $p < 0.01$; ns, not significant). Black line in each subplot represents the mean value of the group, and the dashed line is the mean for the two groups combined. LGM Last Glacial Maximum, MAT mean annual temperature, AP annual precipitation. Pictograms courtesy of Phy-loPic (www.phylopic.org): (**a**) Tracy A. Heath; (**b**) T. Michael Keesey.

higher MAT and AP, and less anomalies in seven out of the eight paleoclimatic variables ($p < 0.001$, Fisher-Pitman permutation test). These findings suggested that these hotspot areas tend to have experienced lower climatic variability over geological time. Furthermore, the three types of hotspots presented varied patterns across the seven tested variables (Figs. S5 and S6). The mixed- and pure paleo-endemism hotspots showed greater likeness to each other, and generally resembled the patterns observed in the combined hotspots (Fig. 3c, d cf. Figs. S5 and S6). Despite sharing many similarities with the other hotspot types when compared to non-hotspots, pure neo-endemism hotspots also contained distinct environmental features. They tended to have smaller MAT, AP, and colder LGM MAT, and higher Miocene AP (for angiosperms) in comparison to other hotspot types ($p < 0.05$), suggesting that neo-endemism hotspots represent areas with relatively unstable climates in the past compared to the other types of PE hotspots. In addition, for gymnosperms, neo-endemism centers demonstrated the lowest MAT, aligning with the facts that most gymnosperm trees are conifers and that conifer diversity tends to peak at mid- rather than low latitudes (Fig. S6a, g).

## Human and future climate threats to tree phylogenetic endemism hotspots

Phylogenetic endemism hotspots are exposed to more pronounced anthropogenic pressures than other areas: these hotspots generally have higher levels of human modification (HMI) values than nonhotspots regions, with gymnosperm-only hotspots being particularly affected ($p < 0.001$, Fisher-Pitman permutation test; Fig. 4a). Furthermore, angiosperm-only hotspots are forecasted to be exposed to substantially higher levels of warming ($p < 0.001$) than all the other three groups, which show no significant differences (Fig. 4b). Additionally, both the separate and joint angiosperm-gymnosperm tree endemism hotspots are anticipated to experience greater future increases in rainfall than non-hotspots ($p < 0.001$, Fig. 4c).

## Current and future protection status of tree phylogenetic endemism hotspots

Existing protected areas have limited protection capacity to tree phylogenetic endemism hotspots. Only 7.4% and 8.7% of angiosperm-only and joint PE hotspots are located in grid cells with existing protected areas (Fig. 5). In addition, no gymnosperm-only hotspots are protected by existing protected areas. Implementing the three conservation prioritization frameworks for tree species diversity[7] would strongly enhance the protection level for all PE hotspots (Fig. 5). By focusing conservation efforts on the top 17% priority areas, substantial increases in protection percentages can be achieved. Specifically, the protection percentage would increase to 67.9%, 36.0%, and 78.6% PE hotspots for angiosperm-only, gymnosperm-only, and joint hotspots encompassing both angiosperm and gymnosperm, respectively. Expanding the conservation scope to include the top 30% priority areas would result in greater protection percentages: 90.4%, 68%, and 92.1% PE hotspots for angiosperm-only, gymnosperm-only, and joint hotspots, respectively. Notably, by safeguarding the top 50% priority areas, these hotspots would be almost entirely protected (≥96%; Fig. 5). Meanwhile, these priority frameworks would strongly increase the protection status of non-hotspot areas from 6.1% (existing protected areas) to 60.9% (top 50% priority areas) as well.

## Discussion

Using global distributions of over 41,000 tree species, we found that high phylogenetic endemism (PE) levels in low- and mid-latitude regions, with these mainly caused by angiosperm species (Fig. 1a). Gymnosperms have high PE in more restricted areas, especially at mid-latitudes plus in the Indo-Malayan-Australasian area (e.g., New Guinea, Tasmania, Japan, and southern China). Both angiosperm and gymnosperm phylogenetic endemism hotspots are associated not only to

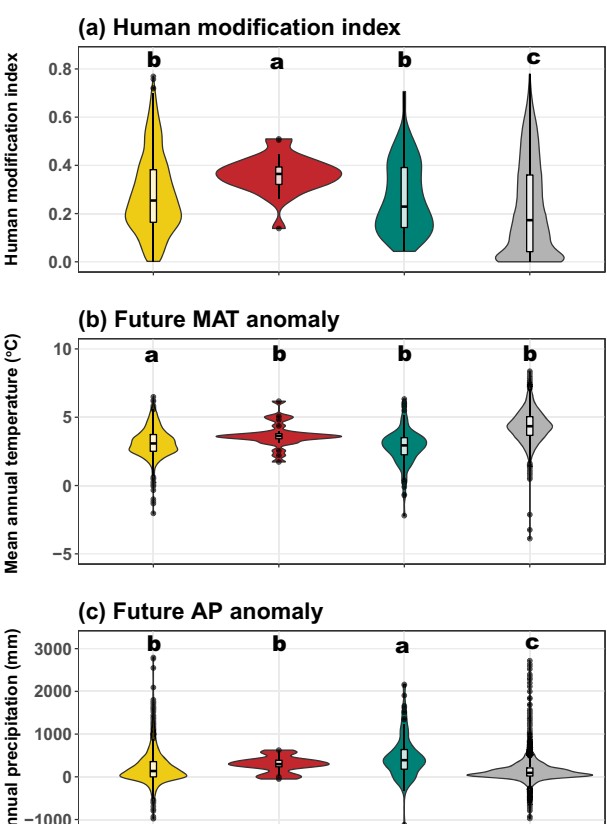

**Fig. 4 | Human threats and future climate change pressure for tree phylogenetic endemic hotspots.** Comparisons of (**a**) current human threats, (**b**) future (2070) mean annual temperature (MAT) anomaly (i.e., future MAT minus present MAT), and (**c**) future (2070) annual precipitation (AP) anomaly for unique phylogenetic endemism hotspots (i.e., centers of neo-, paleo-, and mixed endemism in Fig. 2c) for angiosperms ($n = 1410$) and gymnosperms ($n = 25$) separately, joint hotspots for the two groups ($n = 126$), and non-hotspots (not significant regions in Fig. 2c, $n = 5945$). Significance tests were carried out using the K-sample Fisher-Pitman permutation test (100,000 permutations) and the method "Tukey" for the multiple post-hoc tests. Different letters indicate significant differences among groups ($p < 0.05$ at least). MAT mean annual temperature, AP annual precipitation.

current environmental conditions, but also to long-term climate stability. In addition, tree phylogenetic endemism hotspots are subject to relatively high anthropogenetic and future climate change pressure, and poorly protected. Expanding of global protected areas ambitiously based on the tree diversity conservation prioritization framework[7] would greatly increase their protection level.

The positive associations between current mean annual temperature, annual precipitation and elevation range and angiosperm tree PE (Fig. 3a, b) and significantly elevated temperature, precipitation, and elevation range for PE hotspots (Fig. 3c, d) reflect that high PE regions primarily exist in the mountainous tropics (Figs. 1 and 2). The relatively stable environment there, as indicated by the anomaly variables presented in Figs. 3c, d, S5 and S6, likely have resulted in low extinction rates, leading the tropics both species diversifying[57,58] and persistence centers of tree species, as shown by the presence of large areas of both paleo- and mixed-endemism hotspots (Fig. 2), that is, a combination of ancient lineages that have persisted through time and recently diverging lineages[30,47,48,59,60]. Furthermore, our findings indicate that neo-endemism centers are located in regions experiencing moderate environmental changes, with significantly higher Miocene AP and lower LGM MAT levels compared to the other PE centers, supporting the hypothesis that moderate environmental instability may promote recent specification[30,39,40,47]. In line with the high

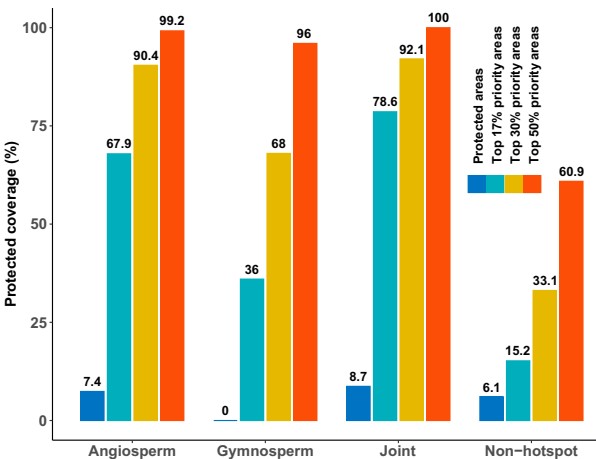

**Fig. 5 | Protection status of tree phylogenetic endemism hotspots and non-hotspots.** The grouped bars and the numbers above represent the protected percentage of phylogenetic endemism hotspots (i.e., centers of neo-, paleo-, and mixed endemism) for angiosperm-only, gymnosperm-only, hotspots for both (joint hotspots), and non-hotspots (not significant regions in Fig. 2c) by existing protected areas, prioritized conservation areas for 17% of Earth's surface target (Top 17% priority areas), the COP15 30×30 target (Top 30% priority areas) and Half-Earth target (Top 50% priority areas), respectively.

proportion of gymnosperm hotspots (128 of 145 cells, or 88.3%) overlapping with angiosperm hotspots (Fig. 2d), the estimated links to the tested environmental variables for gymnosperms are similar to those for angiosperms. In summary, our findings provide support that paleoclimatic change, in conjunction with the current climate, have played a crucial role in shaping both tree PE and the distribution of PE hotspots on a global scale. Importantly, these relations are not solely limited to the Pleistocene glaciations, but extend to deeper time scales (in line with[36,38]), emphasizing the complex interplay between past and present climatic conditions in shaping the current patterns of biodiversity.

In agreement with studies on other organisms[16,22,23,25,48,61], our study revealed that mixed-endemism centers constituted the predominant type of endemism hotspots for both angiosperm (94.1%) and gymnosperm (84.9%) trees. The notable overrepresentation of the mixed-endemism centers reinforces the critical role of stable environments over long periods in both the preservation of ancient lineages and the diversification of new lineages[30,47,48]. For example, the relatively high gymnosperm PE observed in Chile and Tasmania, both of which are mixed-endemism hotspots, offers additional support for the explanation mentioned above. In Chile, the unique environmental conditions, including the diverse range of habitats and the presence of ancient mountain ranges, have likely contributed to the high levels of gymnosperm PE[62,63]. Similarly, Tasmania's complex topography and relatively stable climate across the late Cenozoic may have fostered the evolution and persistence of gymnosperm lineages[48,64,65]. These findings underline the importance of long-term environmental stability in facilitating the coexistence of diverse and evolutionary distinct lineages within these hotspot regions.

Generally, tree PE hotspots face higher levels of threat from human pressures than the non-hotspots (Fig. 4a), although both hotspots and non-hotspots alike receive inadequate protection from the existing protected areas (Fig. 5). Notably, gymnosperm-only hotspots are entirely situated outside of existing protected areas and experience relatively higher levels of human pressures than the other two types of hotspots. These are consistent with previous findings that tree species, particularly those with narrow ranges, are significantly impacted by high levels of human pressure[7], including habitat loss, degradation, and deforestation[66–68]. The alignment strengthens the

evidence for the urgent need to address these human-induced threats and prioritize conservation efforts in tree PE hotspots. It highlights the importance of understanding the specific pressures faced by narrow-ranged tree species (i.e., endemics) and implementing targeted conservation measures to mitigate their impacts, such as the above-mentioned Chile and Tasmania, among the others.

Angiosperm-only PE hotspots or all types of tree PE hotspots will be exposed to either higher temperature or rainfall increases in the future than non-hotspots, respectively. Although increased precipitation may partly alleviate rising warming and associated evaporative demand, they may nevertheless drive changes in vegetation structure and habitat suitability, and the magnitude and rapidity of changes expected within a relatively short timeframe (less than 50 years) in itself poses a risk to the PE hotspots[69], even if all of the identified hotspots are effectively conserved under the prioritized conservation framework that covers the top 50% priority areas. Hence, the potential adverse effects of climate change, featured by altered precipitation regimes and temperatures, intertwined with the ongoing and increasing human activities that result in shifts in habitat suitability, collectively pose important threats to the long-term persistence and ecological functioning of these unique ecosystems. This highlights the need to ensure protected areas are effectively implemented as well as to consider additional actions to help safeguard the hotspots' small-range species representing millions of years of unique evolutionary history, e.g., assisted colonization[70,71] or ex-situ conservation[72].

The identified tree spatial PE hotspots were analogous to those found for other taxonomic groups using the CANAPE approach, e.g., global land vertebrates[23] and Neotropical snakes[16]. The congruence of PE hotspots between tree species and other organisms is likely in part reflect the fundamental functions that trees provide, particularly creating suitable habitats for other organisms[5,27,73,74]. In addition, the same environmental factors could also shape the similar PE patterns between different organisms in the same region, e.g., as also seen between subsets of trees such as Australian eucalypts[29] and acacias[25]. The similarity in PE hotspots for trees to those for other taxonomic groups indicate the utility of tree-targeted protection activities to preserve much of biodiversity and entire ecosystems in many settings (Fig. 5)[7], albeit likely less so in drier and colder biomes, where trees are less diverse or play less important ecological roles.

Improving our understanding of where the phylogenetic rarity of trees is concentrated, how much land use pressure these phylogenetic endemism hotspots are exposed to, and whether they will be maintained under future climate changes is indispensable for efficient conservation of Earth's tree species. We addressed these questions using a comprehensive occurrence dataset and phylogenetic data on 41,835 tree species. Globally, high phylogenetic endemism of tree species occurs mainly at low-to-mid latitudes and significantly associated to areas of long-term climate stability, highlighting that if anthropogenic warming causes such areas to lose their climatic stability this would be a major risk. Concerningly, our findings indicate that these hotspots are particularly exposed to strong changes in climate anticipated in the coming half-century. Moreover, the phylogenetic endemism hotspots are not well covered by existing protected areas and already face high human pressure at present, showing a major risk from land-use-driven ecosystem loss and degradation. Implementing proposed protected area expansion frameworks for prioritizing conservation efforts of tree species' high diversity areas would substantially improve the protection level of not only tree phylogenetic endemism hotspots, but the ecosystem on a whole given the essentially equivalent phylogenetic endemism hotspot patterns in other groups. As these hotspots are of particular importance for biodiversity and ecosystem functioning, their effective conservation and restoration is crucial to reduce the risk to their many endemic species with high evolutionary uniqueness.

## Methods

### Tree species and their range maps

We used the world tree species list[75] and species range maps compiled by[7,54]. The world tree species checklist (GlobalTreeSearch, GTS, v.1.6[75],) was used to extract the global tree species list for the current study. Tree species included in the GTS are based on the definition by the IUCN's Global Tree Specialist Group (GTSG), i.e., "a woody plant with usually a single stem growing to a height of at least two meters, or if multi-stemmed, then at least one vertical stem five centimeters in diameter at breast height"[75]. The Taxonomic Name Resolution Service (TNRS) online tool[76] was used to remove synonyms and to taxonomically standardize the list. The occurrence records of the selected species were collated from five widely used and publicly accessible databases, namely: the Global Biodiversity Information Facility (GBIF; the derived dataset summarizing the used occurrence records from 2070 datasets in GBIF can be viewed and accessed via https://doi.org/10.15468/dd.4jvnmv), the public domain Botanical Information and Ecological Network v.3 (BIEN; http://bien.nceas.ucsb.edu/bien/[14,77]), the Latin American Seasonally Dry Tropical Forest Floristic Network (DRYFLOR; http://www.dryflor.info/[78]), the RAINBIO database (http://rainbio.cesab.org/[79]), and the Atlas of Living Australia (ALA; http://www.ala.org.au/). The compiled occurrence data was assessed[54] and the high-quality records were then used to generate range maps based on the alpha hull algorithm[80] via the *Alphahull* package[81] in R (ver. 4.1.1[82]). We further validated the range maps using three external independent continental and global datasets[30,83–90], and our observed tree diversity pattern fitted well to those obtained from each of the external datasets[7]. The estimated range maps of the 41,835 tree species were rasterized to 110 km equal-area grid cells (-1 degree at the Equator), a resolution commonly used in global diversity studies (e.g ref. [91]), using the *letsR* package[92]. See ref. [7] for detailed information on the range map estimations and three types of external validations. To reduce the potential bias of low-species cells, we only used cells with ≥ five species for further analyses.

### Phylogeny

We constructed a dated phylogenetic tree for tree species using the most comprehensive seed-plant phylogeny (the ALLMB tree[93]). This megatree combines a backbone tree[94], which was built using sequence data from public repositories (GenBank) to reflect deep relationships, with previous knowledge of phylogenetic relationships and species names from the Open Tree of Life (Open Tree of Life synthetic tree release 9.1 and taxonomy version 3, https://tree.opentreeoflife.org/about/synthesis-release/v9.1). As 5,791 species in our 54,020 tree species dataset were missing from the megatree, they were manually added according to its genus or family, a method widely applied in similar studies[7,23,48,93]. We then pruned the phylogeny to contain only species with distribution data (i.e., 41,835). Although the generated phylogeny contains some polytomies, this is unlikely to bias the global analyses of phylogenetic patterns here, as previous study had found that a phylogeny generated by pruning from a synthesis tree has consistent results in community phylogenetic analyses with those based on a purpose-built phylogeny based on gene sequence data[95]. As angiosperm and gymnosperm tree species have remarkably different evolutionary histories and distributions (Fig. 1 and S3)[96], we separated them as two independent groups in the study. Hence, the global tree species phylogeny was divided into two, one for angiosperm species (*n* = 41,275) and one for gymnosperm species (*n* = 560).

### Spatial phylogenetic diversity and endemism analyses

We used the BIODIVERSE software (version 3.1.0)[97] in R using the "Biodiverse pipeline" (https://github.com/NunzioKnerr/biodiverse_pipeline) to perform all the following metric calculations and randomization tests.

Phylogenetic endemism (PE) was calculated as the sum of phylogenetic branch length spanned by species present in each of the 110 × 110 km grid cell, with each branch length divided by the global range size of its descendant clade[13]. Here, PE is scaled to represent the proportion of variation within the tree represented by the given taxa in the grid cell and the total tree length[13,21,97]. We then calculated relative phylogenetic endemism (RPE), which is the ratio of PE measured on the original phylogeny (PE$_{original}$) and estimated from a comparison phylogeny with equally distributed branch lengths (PE$_{equal}$)[25].

We further ran randomization tests to assess grid cells showing more statistical significance than expected, given the species richness and range size per grid cell[25]. The randomization test was achieved by shuffling the terminals of the phylogeny while retaining species richness per grid cell and range size. We ran 999 randomization iterations, and the observed value was compared with the random values. We assigned grid cells as significantly higher or lower than expected as being > 97.5% or <2.5% of the random values, respectively (two-tailed test, α = 0.05). All other grid cells were regarded as not significantly different than expected by chance. Next, PE and RPE were used to classify grid cells relative to the amount of paleo- and neo-endemics according to Categorical Analysis of Neo- and Paleo-endemism (CANAPE analysis)[25]: paleo-endemism was recorded in cells where RPE was significantly high while PE$_{orginal}$ was higher than PE$_{equal}$ in the 97.5% of randomizations; neo-endemism, in contrast, was found in cells where RPE was relatively high while PE$_{original}$ was lower than PE$_{equal}$ in the 97.5% of the randomizations; cells were classified as mixed endemism (mixture of paleo-, and neo-endemism) if RPE was not significantly high or low while both PE$_{original}$ and PE$_{equal}$ were significantly high (97.5% of the randomizations); and cells with neither significantly high PE$_{original}$ nor PE$_{equal}$ were classified as not being endemism centers. Thus, paleo-endemism represented areas with significantly more range-restricted long branches, while neo-endemism were areas with substantially more range-restricted short branches. We then grouped the three endemism types (i.e., paleo-, neo-, and mixed-endemism) as hotspots and non-endemism as non-hotspots for each angiosperm and gymnosperm species, then checked the overlap of PE hotspots between them.

We ran two parallel analyses using a 50 × 50 km resolution and a 220 × 220 km resolution to evaluate the sensitivity of our results (Figs. 7 and S8). However, we only managed to get the gymnosperm analysis at the 50 × 50 km resolution done, as the angiosperm analysis at this resolution was over the computation capacity. The results showed that the spatial endemism hotspots were generally consistent for the three resolutions, and the area ranking of the four types of endemism groups were also largely maintained, with only minimal changes (Fig. 2 *cf*. Fig. S7 and Fig. S8). Given the disparity in grid cell numbers between the resolutions, we consider the varying percentage numbers to be reasonable.

### Environmental variables

To detect the potential drivers of global tree phylogenetic endemism, we compiled a series of environmental variables, including current climate, paleoclimate, and topographic heterogeneity (Table S1). Climate, both present-day and paleoclimate, is generally assumed to be a vital predictor of species distribution and diversity patterns (e.g.,[36,38,39,49,98,99]). We included two bioclimatic predictors commonly used in relevant studies: annual mean temperature (MAT) and annual precipitation (AP). Current climate variables were extracted from the CHELSA database (www.chelsa-climate.org) at a resolution of 30 arc-seconds (-1 km at the Equator)[100], averaging global climate data from the period 1970−2000. We selected two paleo-time periods spanning from *ca*. 11.6−7.2 Mya to *ca*. 21 kya, representing warmer or cooler climatic conditions compared to the present-day climate. Specifically, the late Miocene climate (11.61−7.25 Mya) was used to represent the warmer pre-glacial climate compared to the present day (hereafter

Miocene)[45]. The Last Glacial Maximum (LGM, ~ 21 kya) was used to present the extreme cooling of the Pleistocene glaciations[101]. The LGM data was extracted from the CHELSA database at a resolution of 30 arcseconds[100]. Mean values for all predictors were extracted for each grid cell at a 110 ×110 km resolution. The variable extractions and averaging were carried out in the *letsR* package[92].

In addition to climate, topographic heterogeneity can also affect plant distributions[36,83,102–105] and is considered a universal driver of biological diversities[105,106]. As a proxy of topographic heterogeneity, we computed the elevation range as the absolute difference between the maximum and minimum elevation value within each 110 × 110 km grid cell based on the digital elevation model at 90 m resolution (http://srtm.csi.cgiar.org/).

### Human and future climate threats
We used the human modification index[51] as a proxy of human activities. The human modification index was modelled with 13 most recent global-scale anthropogenic layers (with the median year of 2016) to account for the spatial extent, intensity, and co-occurrence of human activities, many of which show high direct or indirect impact on biodiversity[107]. The index was extracted at 1 km$^2$.

We further extracted future climate variables to calculate future climate anomalies. MAT and AP for 2071–2100 were obtained from the most-updated CHELSA dataset (V2: CMIP6 model, https://chelsa-climate.org/cmip6/)[100]. For CHELSA V2: CMIP6 scenarios, there are only five general circulation models (GCM) available. For each model, three SSP and GCM combinations were provided, i.e., SSP126, SSP370, and SSP 585, reflecting different emission scenarios (Table S2). We downloaded all the 15 scenarios (5 GCMs × 3 SSPs) at a resolution of 30 arc seconds and then aggerated to 110 × 110 km grid cells using either mean or median values. We then averaged the five GCM models for each of MAT and AP, and calculated future climate anomaly as the future MAP/AP minus present MAP/AP. In total, 12 anomaly variables (6 MAT and 6 AP) were obtained. However, as the six MAP /AP anomaly variables are highly correlated (Fig. S9), we selected SSP370 for each MAT and AP to obtain the future MAT and AP anomaly (Figs. S1e, f and S2).

### Existing and potential protected areas
The existing protected areas data were extracted from the November 2021 release of the World Database on Protected areas (WDPA) via the *wdpar* package[108,109]. Following ref. 7, we extracted the protected areas from the WDPA database by selecting terrestrial areas belonging to IUCN protected area categories I to VI and having a status "designated", "inscribed", or "established", and areas not designated as UNESCO Man and Biosphere Reserves, and protected areas represented as points were excluded too. The final extracted protected areas were then resampled at the 110 × 110 km grid cell level.

The three tree diversity priority areas were obtained from ref. 7. Using the same dataset here, we considered tree species' taxonomic, phylogenetic, and functional diversity simultaneously and applied a complementary analysis to identify priority tree species diversity areas for conservation. Specifically, we set three targets of top priority areas: top 17%, top 30%, and top 50%, representing the CBD 2020 target, the COP15 30×30 target, and the Half-Earth proposal[56], respectively.

### Statistical analyses
To represent the amplitude of the climate changes within each time scale, we calculated the anomaly for MAT and AP between the two paleo-time periods and the present-day, i.e., past minus present (Figs. S1a–d and S2)[36,38,39,99,103]. On average, the mean annual temperature (MAT) was much higher in the Miocene and much lower in the LGM compared to the present-day (Fig. S2a). In contrast, the Miocene and LGM had slightly higher and lower precipitation than current precipitation, respectively (Fig. S2b). The paleo-time periods selected,

thus, represent (on average) cold and warm paleoclimates compared to present-day conditions. Pearson correlation coefficients showed low correlations between MAT, AP, and their respective anomaly variables (Fig. S9).

To evaluate the relative importance of the predictor variables in determining the variation in species' phylogenetic endemisms, we used ordinary least squares models (OLSs) and spatial simultaneous autoregressive models (SARs). We tested for predictor collinearity, which could affect our analysis by computing the variance inflation factors (VIFs). All VIFs in each model were smaller than 4, i.e., multicollinearity was not an issue in the analyses. We used the SAR error model because of its superior performance compared to other SAR model types[110]. The SAR error model adds a spatial weights matrix to an OLS regression to account for SAC in the model residuals. A series of spatial weights, i.e., *k*-means neighbor of each site, were tested. Residual SAC was examined in all models (both OLS and SAR) using Moran's *I* test, and Moran's *I* correlograms were also used to visualize the spatial residuals of the models. Model explanatory power was represented by adjusted $R^2$ (OLSs) and Nagelkerke pseudo-$R^2$ (SARs)[111], while the Akaike Information Criterion (AIC) and Bayesian information criterion (BIC) were used to compare the models[112]. SARs and Moran's *I* tests were carried out using the *spdep* and *spatialreg* packages[113,114]. Both OLS and SAR models were run by including current MAT and AP, the four anomaly variables, and the nonclimate predictor, i.e., elevation range, to investigate their relative contributions to each pattern. Before running the models, we inspected the normality of all predictors and log$_{10}$-transformed variables if needed. All response variables were log$_{10}$-transformed. Subsequently, we standardized all predictor variables by transforming all variables to a mean of zero and a standard deviation of one to derive more comparable estimates[115]. We found that SAR models performed better than the corresponding OLS models regarding AIC, BIC, and $R^2$ (Tables S3 and S4), and both SAR models successfully accounted for SAC in model residuals ($p > 0.05$, Fig. S10). Thus, we only represented the results from SARs models in the text, even though the significance of some predictors varied between OLS and SAR models (Fig. S4).

We performed a two-sample Fisher-Pitman permutation test (10,000 permutations) using the *coin* package[116] to compare differences in the variables used in the above models between hotspots and non-hotspot regions for each angiosperm and gymnosperm species. In addition, to represent the human and future climate threats on angiosperm and gymnosperm PE hotspots, k-sample Fisher-Pitman permutation test (10,000 permutations) was carried out, and multiple comparisons were further tested using the *rcompanion* package[117].

### Reporting summary
Further information on research design is available in the Nature Portfolio Reporting Summary linked to this article.

## Data availability
The tree occurrence data are openly available from the TREECHANGE (https://github.com/wyeco/TC_conservation). The data related to analyses are available on Github (https://github.com/wyeco/tree_PE_conservation) and mirrored on Zenodo (https://zenodo.org/record/8418763)[118]. Current and future climate data were extracted from the CHELSA database (https://chelsa-climate.org/); paleoclimate data were obtained from the Paleoclim database (http://www.paleoclim.org) and ref. 45; the human modification index was from ref. 51. The world protected areas were downloaded from https://www.protectedplanet.net/en/thematic-areas/wdpa.

## Code availability
The R codes for the analyses are available on Github (https://github.com/wyeco/tree_PE_conservation) and mirrored on Zenodo (https://zenodo.org/record/8418763)[118].

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

## Acknowledgements

J.-C.-S., W.Y.G., and JMSD acknowledge support from the Danish Council for Independent Research | Natural Sciences (Grant 6108-00078B) to the TREECHANGE project. J.-C.-S. considers this work a contribution to his VILLUM Investigator project "Biodiversity Dynamics in a Changing World", funded by VILLUM FONDEN (grant 16549), and Center for Ecological Dynamics in a Novel Biosphere (ECONOVO), funded by Danish National Research Foundation (grant DNRF173). The BIEN working group was supported by the National Center for Ecological Analysis and Synthesis, a center funded by NSF EF-0553768 at the University of California, Santa Barbara, and the State of California. Additional support for the BIEN working group was provided by iPlant/CyVerse via NSF DBI-0735191. B.J.E. and C.M. were supported by NSF ABI-1565118 and NSF HDR-1934790. B.J.E. was also supported by the Global Environment Facility SPARC project grant (GEF-5810). B.J.E., C.V., and B.S.M. were supported by the Fondation pour la Recherche sur la Biodiversité (FRB) and Electricité de France (EDF) in the context of the CESAB project 'Causes and consequences of functional rarity from local to global scales' (FREE). This work was also conducted as a part of the BIEN Working Group, 2008–2012. We thank all the data contributors and numerous herbaria who have contributed their data to various data compiling organizations for the invaluable data and support provided to BIEN. We thank the New York Botanical Garden; Missouri Botanical Garden; Utrecht Herbarium; the U.N.C. Herbarium; and GBIF, REMIB, and SpeciesLink. The staff at CyVerse provided critical computational assistance. We acknowledge the following herbaria that contributed species occurrence data to the BIEN database (the full names of the herbaria can be found at https://bien.nceas.ucsb.edu/bien/data-contributors/herbaria/): A, AAH, AAH, AAS, AAU, ABD, ABH, ABN, ABRN, ABS, ACAD, ACOR, AD, ADW, AFS, AIMS, AJOU, AK, AKPM, AKU, ALCB, ALTA, ALU, AMD, MADE, AMES, AMNH, AMO, ANA, ANGU, ANSM, ANSP, AQP, ARAN, ARIZ, ARM, AS, ASDM, ASU, ATCC, AU, AUT, AWH, B, BA, BAA, BAB, BABY, BACP, BAF, BAFC, BAI, BAJ, BAK, BAL, BARC, BAS, BASBG, BBB, BBG, BBS, BC, BCB, BCF, BCMEX, BCN, BCRU, BCW, BDD, BDK, BEL, BEREA, BERN, BEX, BFT, BG, BH, BHCB, BHSC, BIO, BIRA, BIRM, BISH, BKF, BLA, BM, BHM, BNRH, BOCH, BOG, BOL, BOLV, BON, BONN, BOON, BORH, BOTU, BOUM, BOZ, BPI, BR, BREM, BRI, BRIST, BRISTM, BRIT, BRIU, BRLU, BRM, BRNU, BRU, BRWK, BRY, BSB, BSIP, BTN, BUF, BUL, BUT, C, CAM, CAMU, CAN, CANB, CAS, CATIE, CAY, CBG, CBM, CBS, CDBI, CEN, CEPEC, CESJ, CGE, CGG, CGMS, CHAM, CHAP, CHAPA, CHAS, CHI, CHL, CHR, CHRB, CHSC, CICY, CIIDIR, CIMI, CINC, CLE, CLEMS, CLF, CLR, CM, CMC, CMM, CMMEX, CNH, CNPO, CNS, COA, COAH, COCA, CODAGEM, COFC, COI, COL, COLO, CONC, COR, CORD, CP, CPAP, CPUN, CR, CRAI, CRP, CS, CSLA, CSU, CTES, CTESN, CU, CUVC, CUZ, CVRD, CYN, DAL, DAO, DAV, DBG, DBN, DCR, DEE, DES, DFS, DFSM, DGS, DHM, DLF, DMFS, DMNH, DMU, DNA, DOR, DPU, DR, DS, DSM, DSY, DUE, DUKE, DUSS, E, EA, EAC, EBH, EBUM, ECON, ECU, EGHB, EIF, EIU, EKY, ELN, EMMA, ENCB, EPM, ER, ERA, ESA, ETH, EXR, F, FAA, FAU, FAUC, FB, FCME, FCO, FCQ, FEN, FH, FHO, FI, FLAS, FLOR, FM, FR, FRS, FRU, FSC, FSU, FTG, FUEL, FULD, FURB, G, GA, GAT, GB, GDA, GENT, GEO, GES, GGO, GH, GI, GJO, GL, GLAM, GLM, GMDRC, GMNHJ, GOET, GOW, GRA, GRM, GSW, GUA, GUAM, GW, GZU, H, HA, HAC, HAL, HAM, HAMAB, HAMU, HAO, HAS, HAST, HASU, HAW, HB, HBG, HBR, HCIB, HDD, HEID, HGM, HIB, HIP, HITBC, HIWNT, HLU, HME, HNT, HNUB, HO, HPL, HRCB, HRP, HSS, HSU, HTN, HU, HUA, HUAA, HUAL, HUAZ,

HUCP, HUEFS, HUEM, HUFU, HUJ, HUSA, HUT, HWB, HXBH, HYO, IAA, IAC, IAN, IB, IBF, IBGE, IBK, IBSC, IBUG, ICEL, ICESI, ICN, IEA, IEB, IFO, ILL, ILLS, IMSSM, INB, INEGI, INIF, INM, INPA, INV, IPA, IPRN, ISC, ISE, ISKW, ISL, ISTC, ISU, IZAC, IZTA, JACA, JBAG, JBGP, JCT, JE, JEPS, JUA, JYV, K, KAND, KATH, KCS, KGY, KHD, KIEL, KLE, KMN, KMNH, KOELN, KOR, KPABG, KPM, KR, KSC, KSP, KSTC, KSU, KTU, KU, KUN, KYO, L, LA, LAE, LAGU, LAM, LANC, LAU, LBG, LCN, LCR, LD, LDS, LE, LEA, LEB, LEI, LES, LI, LIL, LINN, LISC, LISE, LISI, LISU, LIV, LIVU, LL, LLN, LMA, LMG, LMU, LOB, LOJA, LOMA, LP, LPAG, LPB, LPD, LPS, LSR, LSU, LSUM, LTB, LTR, LU, LUA, LW, LYJB, LZ, M, MA, MACB, MACF, MAF, MAK, MAL, MAN, MANCH, MARY, MASS, MB, MBA, MBH, MBK, MBM, MBML, MCNS, MEL, MELU, MEN, MERL, MEXU, MFA, MFU, MG, MGC, MHA, MICH, MIL, MIN, MISS, MJG, MMMN, MNA, MNE, MNHM, MNHN, MO, MOL, MOR, MOSS, MPU, MPUC, MSB, MSC, MSTR, MSUB, MSUN, MT, MTMG, MU, MUB, MUCV, MUR, MVFA, MVFQ, MVJB, MVM, MW, MY, N, NA, NAC, NAS, NCC, NCCBH, NCCE, MCCH, NCE, NCH, NCSC, NCU, ND, NE, NEBC, NH, NHA, NHG, NHM, NHMC, NHT, NM, NMB, NMLU, NMNL, NMR, NMSU, NOT, NOU, NSPM, NSW, NT, NTS, NU, NUM, NWOSU, NY, O, OBI, OCLA, ODU, OKL, OKLA, OS, OSA, OSC, OSH, OULU, OWU, OXF, P, PACA, PAMP, PAR, PCU, PDD, PE, PEL, PERTH, PEUFR, PFC, PGM, PH, PI, PKDC, PLAT, PMA, PMSP, PNG, PNH, POLL, POM, PORT, PR, PRC, PRE, PSU, PSY, PTH, PVNH, PY, QCA, QCNE, QFA, QK, QM, QRS, QUE, R, RAMM, RAS, RB, RBR, RDG, RDS, REG, RELC, RENO, RFA, RIOC, RM, RNG, RSA, RYU, S, SALA, SAM, SAN, SANT, SAPS, SASK, SAV, SBBG, SBT, SCAR, SCFS, SD, SDN, SDSU, SEL, SEV, SF, SFD, SFS, SFV, SGO, SHD, SHIN, SHYB, SI, SIM, SING, SIU, SJC, SJRP, SJSU, SLBI, SLPM, SLSK, SMDB, SMF, SNM, SNUA, SOM, SP, SPF, SPN, SPSF, SQF, SRFA, SRGH, SSMF, STA, STDCM, STE, STI, STL, STR, STS, STU, SUN, SUU, SUVA, SVG, SWT, SZU, TAA, TAAM, TAES, TAI, TAIF, TALL, TAM, TAMU, TAN, TASH, TBI, TCD, TEF, TENN, TEPB, TEX, TFC, TI, TKPM, TNP, TNS, TO, TOGO, TOR, TOYA, TRA, TRH, TROM, TRT, TRTE, TRTS, TTN, TUB, TUCH, TULS, U, UADY, UAM, UAMIZ, UB, UBA, UBC, UBT, UC, UCI, UCMM, UCNM, UCR, UCS, UCSA, UCSB, UCSC, UEA, UEC, UESC, UFG, UFMA, UFMT, UFP, UFRJ, UFRN, UFS, UGDA, UH, UI, UJAT, ULM, ULS, UME, UMO, UNA, UNB, UNCC, UNEX, UNL, UNM, UNR, UNSL, UNSW, UPCB, UPEI, UPNA, UPP, UPS, US, USAS, USC, USCH, USF, USJ, USM, USNC, USP, USZ, UT, UTC, UTEP, UU, UVIC, UWO, V, VAL, VALD, VALPL, VDB, VEN, VIT, VMSL, VPI, VT, W, WAG, WAR, WAT, WB, WCR, WCUH, WELT, WFU, WHN, WIES, WII, WIN, WIS, WM, WMNH, WOLL, WOS, WS, WSCO, WSY, WTU, WU, WXM, XAL, YA, YAMA, YK, YRK, Z, ZCM, ZMT, ZSS, and ZT. A complete list of data contributors to the BIEN database can be found at https://bien.nceas.ucsb.edu/bien/data-contributors/all/.

## Author contributions

W.-Y.G. conceived the idea and designed the study with refinement by J.-C.S.; J.M.S.-D., W.L.E., B.S.M., C.M., C.V., M.J.P., M.S., F.S., A.B.-O., B.J.E. helped with data collection; W.-Y.G. performed the analyses and wrote the manuscript with significant contributions from J.M.S.-D. and J.-C.S. All authors revised the manuscript and approved the final submitted version.

## Competing interests

The authors declare no competing interests.
