## [Peer Review File · Nature Communications]

Climate change and land use threaten global hotspots of phylogenetic endemism for treesREVIEWER COMMENTS

Reviewer #1 (Remarks to the Author):

In this manuscript, entitled "Climate change and land use threaten global hotspots of phylogenetic endemism for trees", the authors map the geographical distribution of tree phylogenetic endemism (PE) with the objectives of exploring: i) the macroecological drivers of their spatial distribution, ii) their current conservation status, and iii) their exposure to future climatic and anthropogenic pressures. The topic addressed in the manuscript is very interesting and current, particularly because it provides criteria for setting priorities in addressing the biodiversity extinction crisis. The focus on the spatial distribution of threatened evolutionary singularities provides a global perspective on the risk of tree extinction.

The stated objectives are well addressed and provide clear evidence on the characteristics of tree PE hotspots. The only objective that sounds somewhat confusing and whose results and interpretations may require some rewording is the one on the importance of climate stability in the geographic distribution of PE hotspots.

Minor comments

- Discussing the high PE of gymnosperms in Chile and Tasmania (Fig. 1) would be useful for both understanding the evolutionary legacy of southern hemisphere biotas and for local-scale conservation plans.

Line 128 - "the Andean region of South America" (rather than "the Coastal regions") best describes the distribution of mixed endemics for Angiosperms in south America.

Line 161-163 - following the results Figs. 3A and 3B, and tables S3 and S4 it appears that MAT is the strongest factor with positive relationships with PE for Angiosperms.

Line 234-236. It is still not clear to me how the significant relationship between the LGM AP anomaly and MAT with PE endemism means stability. If you can reword it to make it clearer, that would be great.

Line 251-253. MAT is not significantly associated with gymnosperm PE hotspots.

Reviewer #2 (Remarks to the Author):

Dear Authors,

I enjoyed reading your paper and I am fascinated by your findings. Here are my thoughts and suggestions for minor reviews of this paper, for your consideration:

Line 52: please replace the cited paper with reference number 13

Line 134 - remove the duplicated "in the mountainous regions"

Fig. S3 - It seems like there is major bias in the occurrence/richness and sequenced genetic data for tropical trees in Africa and Asia. While it can potentially skew the results of this study, it also highlights relative neglect of scientific work (particularly DNA sequences) for these regions, in spite of the biodiversity they hold.

Line 276-280: habitat loss, the greatest threat to biodiversity, is countered by habitat protection, that is, creation of protected areas. However, this does not address threats posed by climate change. Moreover, we need studies to show the types and magnitudes of responses of trees to climate change. Some may migrate or adapt (including phenology shifts, etc.), and others might go extinct, but then opens the door for occupation of empty niches by newly formed or migrating

species. Personally, I do not see strong and logical argument on how PE would be affected by climate change in this study. To strengthen the argument, the functional traits of these trees need to be included in the analysis.

Related to this, the entire Discussion is based on climate-PE coupling, with little to no explanation on the human modification index. If tropical ecosystems will experience warmer and wetter climates, then the impact of climate change on these systems will be minimal compared to temperate ecosystems. However, tropical areas face higher population growth and dependence on natural resources, and I would expect some focus on how this will impact future PE patterns in the Discussion.

Line 338-339: the justification of using 110x110 km resolution based on the choice made in previous studies may be outdated. Those previous studies may have encountered computational limitation, which may not be a limiting factor for your research. It is important to, at least, run geospatial analyses like this in different resolutions to decide an appropriate one. Sensitivity analyses could be conducted to check for difference in results.

Line 342-354 – the authors mapped centers of endemism based on CANAPE but it is unclear if a chronogram tree was used for the analysis. If I remember correctly, the Smith and Brown (2018) tree used in this study is a chronogram. However, it should be specified in this paper. If perchance, the tree is not a chronogram, then this paper cannot classify areas of paleo- and neo-endemic taxa. If a phylogram was used, then endemism centers reflect grid cells with significant concentrations of range restricted taxa with short or long accumulated evolutionary changes, and not evolutionary age that is needed to determine neo- and paleo-endemism. Chronogram cannot be used interchangeably with phylogram, since they reflect different biological attributes. Moreover, studies have shown that chronogram and phylogram will give you different results of biodiversity concentrations (see Elliott et al. 2018, *J. Biogeog.*; Kling et al 2019, *Phil. Trans. Royal Soc. B*).

Line 351-352: I love the idea of partitioning the angiosperm and gymnosperm phylogenies based on varying evolutionary histories

Line 359-362: Understandably, the analyses in this paper are based on PE, but the methods also show that PD was calculated, and a good portion of the methods section was dedicated to explaining PD calculations, but PD results were omitted in the paper. I would recommend including them in the supplementary information.

Line 370-371 – PD correlates strongly with species richness, but not so with PE. The findings of this study also show clear incongruence between PE and species richness patterns. And I could not find the place in the paper where this was “mentioned earlier” as used in this sentence. Please revise.

Line 418: would the choice to extract data associated with human modification index in a different resolution (1 km) not affect the results of the study? Secondly, is this anthropogenic activity proxy layer better than human footprint index?

Line 441-442 – in my opinion, the half-earth proposal is unrealistic, given the human modification of the earth since prehistory. This proposal is incongruent with literature showing that 85% of the world’s plant diversity is located in roughly one-third of the land surface. Moreover, this proposal is not sensitive to the impact it will have on the poor people living in developing economies (See Pimm et al. 2018, *Science*). I would have expected the CBD’s 30x30 to be considered here, rather than the half-earth proposal.

The drivers of floristic diversification vary for the global tropical regions. For example, the Amazon Basin enjoyed the Great American Biotic Interchange, which the remaining tropical regions did not experience. Could these unique biogeographical events have impacted the PD and PE patterns?

Reviewer #3 (Remarks to the Author):

Review NCOMMS-23-13110-T

Guo et al. used a large dataset of global tree distribution to identify areas of phylogenetic endemism. They studied the relationship of phylogenetic endemism with present climate as well as climate change since the Late Miocene and the Last Glacial Maximum, respectively. They also looked at whether hotspots of phylogenetic endemism are affected by human land modification and protection at present and if they will experience strong climatic change under future change scenarios. They identified both areas of paleo- and neoendemism, which mostly overlapped between angiosperms and gymnosperms. Current precipitation was most strongly correlated with phylogenetic endemism among climate variables. Importantly, the authors found that areas of high phylogenetic endemism were poorly covered by existing protected areas but strongly impacted by human land use. Climate change, especially increased rainfall, will also disproportionately affect hotspots of tree phylogenetic endemism.

The study is original and significant in (i) studying a key functional group of organisms, which as the authors note in the discussion, can serve as a guide for conserving biodiversity more generally; (ii) employing a large dataset and asking relevant and clear questions regarding past, present and future of tree diversity.

The authors use appropriate statistical methods based on established metrics of phylogenetic endemism and spatial regression models.

The conclusions are valid given the analyses.

The text is well structured and easy to follow; some copy-editing is necessary and some figures could be simplified (see below).

References are appropriate although the authors may want to consider Vasconcelos et al. (2022), see below, and the original Stebbins reference discussed therein, for their "cradles" and "museums" metaphors.

General comments

The abstract mentions very broadly "coupling not just to current climate, but also to paleo-climatic stability" as results. This could be more detailed, citing some of the specific effects found.

In the abstract and introduction, the authors seem to confound rarity and endemism. They should probably stick with "endemism" throughout for clarity, although they could mention that endemism is strongly correlated with rarity.

The authors mention "cradles" and "museums" of biodiversity both in the introduction and the discussion. These metaphors introduced by Stebbins is often used, but the authors may want to consult the excellent review of Vasconcelos et al. (2022, DOI: 10.1086/717412) to get these metaphors right, if they choose to keep them.

The authors introduce two opposing hypotheses on the effect of climatic stability on diversification in lines 82 and 87. The text would benefit if these hypotheses (with potentially their consequences for conservation) were framed more explicitly. Again, Vasconcelos et al. 2022 could help here.

Maybe I am missing this, but are the calculated PE values available as dataset? This may be useful for conservation planners.

Could the authors make the data and code available in a permanent repository (e.g. Zenodo), not just GitHub?

Specific things noticed in passing

L 34: evolutionary -> evolutionarily

Fig 1: resolution may need to be increased

Fig 3: The arrows are a bit confusing und unnecessary. Maybe best to remove them, as the violin and box plots themselves already show the relative difference.

L 183: significant -> significance

Fig 4, small letters: font sizes really difficult to interpret. Why not just add the size of each group above or below the violins?

Fig. 5: Doughnuts really not easy to interpret, especially for the gymnosperms. A series of simple bar plots would perhaps work best here.

L 330 and onwards: Several regional distribution datasets were used. Is there more uncertainty around mapped distribution in areas not covered by these datasets, and could you quantify this?

L 361: matrices -> probably "metrics" intended here?

L 405: Where was the Late Miocene data extracted from? Also from CHELSA?

signed - Jan Hackel

REVIEWER COMMENTS

Reviewer #1 (Remarks to the Author):

In this manuscript, entitled "Climate change and land use threaten global hotspots of phylogenetic endemism for trees", the authors map the geographical distribution of tree phylogenetic endemism (PE) with the objectives of exploring: i) the macroecological drivers of their spatial distribution, ii) their current conservation status, and iii) their exposure to future climatic and anthropogenic pressures. The topic addressed in the manuscript is very interesting and current, particularly because it provides criteria for setting priorities in addressing the biodiversity extinction crisis. The focus on the spatial distribution of threatened evolutionary singularities provides a global perspective on the risk of tree extinction.

The stated objectives are well addressed and provide clear evidence on the characteristics of tree PE hotspots. The only objective that sounds somewhat confusing and whose results and interpretations may require some rewording is the one on the importance of climate stability in the geographic distribution of PE hotspots.

RE: Thank you for the positive assessment of our work. We have carefully revised the sections that discuss the phylogenetic endemism (PE) hotspots and their potential explanation, taking into account the valuable insights provided by the paper suggested by Reviewer 3. Specifically, to enhance clarity and improve understanding, we have rephrased the text introducing the climate stability theory “(Lines 91-102) *Regions characterized by long-term climatic stability over geological time scales, coupled with optimal conditions for plant growth, have played a significant role in driving high rates of speciation and low rates of extinction. These conditions, in turn, increase the likelihood of immigration, resulting in higher levels of PE and the presence of either paleo-endemism or neo-endemism centers. Conversely, regions that have experienced pronounced climate oscillations, such as those occurring during glacial-interglacial cycles, are expected to exhibit both high speciation and extinction rates. This dynamic leads to substantial species turnover, and reduced chance for immigration, thereby could contribute to the formation of neo-endemism centers. Up to now, an explicit test of these hypotheses for global tree PE centers is missing, limiting our understanding of the vulnerability of tree PE hotspots, representing long-term importance for tree survival and diversification, to current human and future climate change threats^{52,53}, and from delineating efficient conservation planning.*”, making the arguments more explicit and accessible. In the Discussion section, we have extensively rewritten the two paragraphs (Lines 267-300) pertaining to this result. We hope the reviewer will find that our revised version has been greatly strengthened.

Minor comments

- Discussing the high PE of gymnosperms in Chile and Tasmania (Fig. 1) would be useful for both understanding the evolutionary legacy of southern hemisphere biotas and for local-scale conservation plans.

RE: Thank you for bringing this to our attention. We agree that the information regarding the relatively high gymnosperm PE in Chile and Tasmania, both identified as mixed-endemism hotspots, is valuable for the discussion. We have incorporated this information into the discussion, specifically in the section discussing the association between gymnosperm PE and paleoclimate. By highlighting these two hotspots and their significance, we aim to provide additional support to our hypothesis regarding the role of stable environments in shaping gymnosperm PE hotspots. Now it reads “(Lines 292-300) *the relatively high gymnosperm PE observed in Chile and Tasmania, both of which are mixed-endemism hotspots, offers additional support for the explanation mentioned above. In Chile, the unique environmental conditions, including the diverse range of habitats and the presence of ancient mountain ranges, have likely contributed to the high levels of gymnosperm PE. Similarly, Tasmania’s isolation as an island and its complex topography may have fostered the evolution and persistence of distinct gymnosperm lineages. These findings underline the importance of long-term environmental stability in facilitating the coexistence of diverse and evolutionary distinct lineages within these hotspot regions.*”

Line 128 - "the Andean region of South America" (rather than "the Coastal regions") best describes the distribution of mixed endemics for Angiosperms in south America.

RE: Changed as suggested.

Line 161-163 - following the results Figs. 3A and 3B, and tables S3 and S4 it appears that MAT is the strongest factor with positive relationships with PE for Angiosperms.

RE: Thanks for pointing this out. We have carefully checked the results and these sentences had been revised as "(Lines 175-178) *Present-day annual precipitation (AP) emerged as a dominant factor, with either the strongest or the second strongest standardized effect, with a positive relation to PE for both gymnosperms and angiosperms (Fig. 3A & 3B). However, present-day mean annual temperature (MAT) had an even stronger, positive association with PE in the case of angiosperms.*"

Line 234-236. It is still not clear to me how the significant relationship between the LGM AP anomaly and MAT with PE endemism means stability. If you can reword it to make it clearer, that would be great.

RE: As mentioned previously, we have reworded the text to make clear the relationships between anomaly and endemism "(Lines 179-185) *With respect to the paleoclimatic variables, LGM AP and MAT anomaly (i.e., LGM AP/MAT minus present AP/MAT) showed consistent positive relations to PE for both groups, indicating that high PE is associated to relatively warm and wet LGM conditions. Both Miocene anomalies show no relation to gymnosperm PE. On the other hand, Miocene MAT and AP anomaly exhibited positive and negative associations with angiosperm PE, respectively, indicating warmer regions with less precipitation during the Miocene than at present have higher angiosperm PE.*"

In addition, we replotted Fig. 3C & 3D using the absolute values to show the differences between endemism hotspots and non-hotspots for the tested environmental variables, and described the relevant results as "(Lines 186-200) *Comparing the combined endemism hotspots (i.e., neo-, paleo-, and mixed endemism centers, as shown in Fig. 2A & 2B) to non-hotspots (not significant regions in Fig. 2A & 2B), both angiosperm (Fig. 3C) and gymnosperm (Fig. 3D) hotspots exhibited greater elevation ranges, higher MAT and AP, and less anomalies in seven out of the eight paleoclimatic variables ($p < 0.001$, Fisher-Pitman permutation test). These findings suggested that these hotspot areas tend to have experienced lower climatic variability over geological time. Furthermore, the three types of hotspots presented varied patterns across the seven tested variables (Figs. S5 & S6). The mixed- and paleo-endemism hotspots showed greater likeness to each other, and generally resembled the patterns observed in the combined hotspots (Figs. 3C & 3D cf. Figs. S5 & S6). Despite sharing many similarities as the other hotspots types when compared to non-hotspots, neo-endemism hotspots were exposed to distinct environmental features. They tended to have smaller MAT, AP, and colder LGM MAT, and higher Miocene AP (for angiosperms) in comparison to other hotspot types ($p < 0.05$), suggesting that neo-endemism hotspots represent areas with relatively unstable climates in the past. In addition, for gymnosperms, neo-endemism centers demonstrated the*

lowest MAT, which aligns with the fact that the majority of gymnosperm trees are conifers (Fig. S6A & S6G)."

Line 251-253. MAT is not significantly associated with gymnosperm PE hotspots.

RE: We added "angiosperm" here (Line 268), as gymnosperm trees generally have low diversity in the tropics.

Reviewer #2 (Remarks to the Author):

Dear Authors,

I enjoyed reading your paper and I am fascinated by your findings. Here are my thoughts and suggestions for minor reviews of this paper, for your consideration:

RE: Thank you for your positive comments to our study.

Line 52: please replace the cited paper with reference number 13

RE: Corrected as suggested.

Line 134 – remove the duplicated "in the mountainous regions"

RE: Thanks for pointing this out and we had corrected as suggested.

Fig. S3 – It seems like there is major bias in the occurrence/richness and sequenced genetic data for tropical trees in Africa and Asia. While it can potentially skew the results of this study, it also highlights relative neglect of scientific work (particularly DNA sequences) for these regions, in spite of the biodiversity they hold.

RE: Even though our dataset is probably the most comprehensive one for tree species globally, we are aware of the potential bias of data deficiency as the reviewer said. Indeed, the data coverage in tropical Africa and Asia was not as extensive as other regions, particularly North America and many parts of Europe. However, not all the countries in these two regions (Africa and Asia) are significantly lacking phylogenetic knowledge (see Rudbeck et al., 2022). Even though some species are missing from these regions, but their evolution histories (i.e., branch length) can be represented by species or their sisters occurred in other regions, as for species for which there was no genetic data available, they were added manually based on current phylogenetic knowledge according to the same method applied by Smith & Brown (2018). This was described in details in Lines 387-396.

Moreover, in previous studies using the same dataset, such as Guo et al. (2022, PNAS) and Xu et al. (2023, Science Advances), we carried out several thorough external validations to check the quality of the distribution data. Briefly, we used three different external datasets, i.e., 1) two continental-scale tree species ranges (EU-Forest (Mauri et al., 2017) and Little's "Atlas of United States trees" (Little, 1971, 1976, 1977, 1978)); 2) a model-generated global tree species richness map (Keil & Chase 2019); and 3) a sub-national level tree species richness map (Sandel et al., 2020). All the three external

validations verified the robustness of our global tree species distributions. While the data is not perfect, we believe that the diversity patterns found here are consistent to establish relationship with paleoclimate.

We modified the relevant sentences in the Methods section to emphasize that data validations had been carried out previously “(Line 383) *See ref. 7 for detailed information on the range map estimations and three types of external validations.*”

Papers cited:

- Guo WY, Serra-Diaz JM, Schrod F, et al. High exposure of global tree diversity to human pressure. *Proc. Natl. Acad. Sci.* 119, e2026733119 (2022).
- Keil P & Chase J M. Global patterns and drivers of tree diversity integrated across a continuum of spatial grains. *Nat. Ecol. Evol.* 3, 390–399 (2019).
- Little, EL, Jr. 1971. Atlas of United States trees. Volume 1. Conifers and important hardwoods. Misc. Publ. 1146. Washington, DC: U.S. Department of Agriculture, Forest Service.
- Little, EL, Jr. 1976. Atlas of United States trees. Volume 3. Minor western hardwoods. Misc. Publ. 1314. Washington, DC: U.S. Department of Agriculture, Forest Service.
- Little, EL, Jr. 1977. Atlas of United States trees. Volume 4. Minor eastern hardwoods. Misc. Pub. No. 1342. Washington, DC: U.S. Department of Agriculture, Forest Service.
- Little, EL, Jr. 1978. Atlas of United States trees. Volume 5. Florida. Misc. Publ. 1361. Washington, DC: U.S. Department of Agriculture, Forest Service.
- Mauri A, Strona G & San-Miguel-Ayanz J. EU-Forest, a high-resolution tree occurrence dataset for Europe. *Sci. Data.* 4, 160123 (2017).
- Rudbeck AV, Sun M, Tietje M, et al. The Darwinian shortfall in plants: phylogenetic knowledge is driven by range size. *Ecography* 2022, e06142 (2022).
- Sandel B et al. Current climate, isolation and history drive global patterns of tree phylogenetic endemism. *Glob. Ecol. Biogeogr.* 29, 4–15 (2020).
- Smith SA. & Brown JW. Constructing a broadly inclusive seed plant phylogeny. *Am. J. Bot.* 105, 302–314 (2018).
- Xu WB, Guo WY, Serra-Diaz JM, et al. Global beta-diversity of angiosperm trees is shaped by Quaternary climate change. *Sci. Adv.* 9, eadd8553 (2023).

Line 276-280: habitat loss, the greatest threat to biodiversity, is countered by habitat protection, that is, creation of protected areas. However, this does not address threats posed by climate change. Moreover, we need studies to show the types and magnitudes of responses of trees to climate change. Some may migrate or adapt (including phenology shifts, etc.), and others might go extinct, but then opens the door for occupation of empty niches by newly formed or migrating species. Personally, I do not see strong and logical argument on how PE would be affected by climate change in this study. To strengthen the argument, the functional traits of these trees need to be included in the analysis.

Related to this, the entire Discussion is based on climate-PE coupling, with little to no explanation on the human modification index. If tropical ecosystems will experience warmer and wetter climates, then the impact of climate change on these systems will be

minimal compared to temperate ecosystems. However, tropical areas face higher population growth and dependence on natural resources, and I would expect some focus on how this will impact future PE patterns in the Discussion.

RE: We sincerely appreciate the thorough comments provided by the reviewer regarding the effects of human activities on tree diversity and PE. We acknowledge the significance of habitat loss as a key factor influencing tree diversity, and we agree with the reviewer that incorporating functional traits could further enhance our understanding of biodiversity changes. However, we believe that the inclusion of functional aspects may deviate from the primary focus of our study and potentially disrupt the logical flow of our research. Moreover, detailed functional data were not available for our analysis at this time.

In response to the reviewer's feedback, we have expanded the discussion to address the potential impacts of human activities on tree PE. We have also attempted to establish connections between human pressures and climate change "(Lines 302-312) *Generally, tree PE hotspots face higher levels of threat from human pressures than the non-hotspots (Fig. 4A), although both hotspots and non-hotspots alike receive inadequate protection from the existing protected areas (Fig. 5). Notably, gymnosperm-only hotspots are entirely situated outside of existing protected areas and experience relatively higher levels of human pressures than the other two types of hotspots. These are consistent with previous findings that tree species, particularly those with narrow ranges, are significantly impacted by high levels of human pressure, including habitat loss, degradation, and deforestation. The alignment strengthens the evidence for the urgent need to address these human-induced threats and prioritize conservation efforts in tree PE hotspots. It highlights the importance of understanding the specific pressures faced by narrow-ranged tree species (i.e., endemics) and implementing targeted conservation measures to mitigate their impacts, such as the above-mentioned Chile and Tasmania, among the others.*"

As mentioned in our response to Reviewer 1, we have thoroughly revised the sections discussing the association between climate and PE (Lines 91-102 & 267-300). We hope the reviewer will find the clarity and coherence in the revised version has been greatly improved.

Line 338-339: the justification of using 110x110 km resolution based on the choice made in previous studies may be outdated. Those previous studies may have encountered computational limitation, which may not be a limiting factor for your research. It is important to, at least, run geospatial analyses like this in different resolutions to decide an appropriate one. Sensitivity analyses could be conducted to check for difference in results.

RE: We acknowledged the potential impact of resolution on our study's results, and we shared the same caution as the reviewer. Our decision to utilize the 110 x 110 km resolution was based on its widespread use in global-scale studies, as it strikes a favorable balance between precision and efficiency. However, during the revision process, we attempted to conduct a parallel analysis using a 50 x 50 km resolution.

Regrettably, we encountered challenges when processing the angiosperm data. Under the 100 x 100 km resolution, we initially had 1,058,003 records (used to prepare the input data for the main analysis), whereas under the 50 x 50 km resolution, we had 4,722,132 records. This represented a 4.5-fold increase in data, causing computation interruptions even with our most powerful server at Aarhus University (processor: Intel Xeon CPU E5-2643 v3 @ 3.40 GHz; RAM: 384GB).

Regarding the gymnosperm data, where we could study how different resolution affects the results, we found that the spatial endemism hotspots were consistent between 50 x 50 km and 100 x 100 km resolutions. The area ranking of the four types of endemism groups was also maintained, with only minimal changes in 0.45%, 2.97% and 0.30% for Paleo, Mixed and Neo endemism hotspots, respectively (see Fig. R1, the two inserted pie graphs). Given the disparity in cell numbers between the resolutions (5,657 vs. 1,200), we consider the varying percentage numbers to be reasonable. Furthermore, we observed that the significance of the tested variables in explaining gymnosperm PE remained the same for both resolutions (see Fig. R2 vs. Fig 3b).

Fig. R1 Comparison of global distribution of gymnosperm tree endemism types. (A) and (B) each represents the result using a 50 x 50 km and a 110 x 110 km resolution (the one used in the main text). Centers of neo-endemism (Neo, i.e., concentrations of rare short branches), paleo-endemism (Paleo, i.e., concentrations of rare long branches), and mixed neo- and paleo-endemism (Mixed) were identified using a randomization analysis.

Fig. R2 Determinants of global phylogenetic endemism (PE) in gymnosperm tree species under the 50 x 50 km resolution. Estimates (standardized slopes) and 95% confidence intervals (C.I.s) were obtained from the (A) linear regression and (b) best spatial auto-regressive model.

Despite being unable to conduct parallel analyses for both groups of trees, our gymnosperm analysis support the suitability of the current 110 x 110 km resolution for our study. In addition, we ran the main analysis using a coarser resolution of 220 x 220 km (Fig. R3). Similarly, the general patterns of the endemism centers kept. These results should be robust enough, particularly for the specific topic we have discussed. We have included those additional analyses in the methods “(Lines 430-437) We ran two parallel analyses using a 50 x 50 km resolution and a 220 x 220 km resolution to evaluate the sensitivity of our results (Figs. 7 & S8). However, we only managed to get the gymnosperm analysis at the 50 x 50 km resolution done, as the angiosperm analysis at this resolution was over the computation capacity. The results showed that the spatial endemism hotspots were generally consistent for the three resolutions, and the area ranking of the four types of endemism groups were also largely maintained, with only minimal changes (Fig. 2 cf. Fig. S7 and Fig. S8). Given the disparity in grid cell numbers between the resolutions, we consider the varying percentage numbers to be reasonable.” and the supporting information (Figs. S7 & S8). We sincerely appreciate the constructive comment provided by the reviewer, as it has bolstered our confidence in the findings.

Fig. R3 Global distribution of tree endemism types for (A) angiosperm and (B) gymnosperm tree species, and (C) their overlap, based on a 220 x 220 km resolution. Centers of neo-endemism (Neo, i.e., concentrations of rare short branches), paleo-endemism (Paleo, i.e., concentrations of rare long branches), and mixed neo- and paleo-endemism (Mixed) were identified using a randomization analysis [categorical analysis of neo- and paleo-endemism (CANAPE)] conducted separately for angiosperm and gymnosperm trees. (D) Venn diagram showing the area and percentage overlap of the significant endemism regions (i.e., hotspot, including centers of either neo-, paleo- or mixed neo- and paleo-endemism) between angiosperm and gymnosperm tree species. Numbers are sum of the hotspot cells, and the percentages in brackets correspond to the percentages of non-overlapping cells in each of the angiosperm and gymnosperm hotspots.

Line 342-354 – the authors mapped centers of endemism based on CANAPE but it is unclear if a chronogram tree was used for the analysis. If I remember correctly, the Smith and Brown (2018) tree used in this study is a chronogram. However, it should be specified in this paper. If perchance, the tree is not a chronogram, then this paper cannot classify areas of paleo- and neo-endemic taxa. If a phylogram was used, then endemism centers reflect grid cells with significant concentrations of range restricted taxa with short or long accumulated evolutionary changes, and not evolutionary age that is needed to determine neo- and paleo-endemism. Chronogram cannot be used interchangeably with phylogram, since they reflect different biological attributes. Moreover, studies have shown that chronogram and phylogram will give you different results of biodiversity concentrations (see Elliott et al. 2018, J. Biogeog.; Kling et al 2019, Phil. Trans. Royal Soc. B).

RE: Thanks for pointing this out and the detailed description. As the reviewer said, the phylogeny we built was based on the megatree of Smith & Brown (2018), which is indeed a chronogram. We added this information in the revision "(Lines 387-388) *We constructed a dated phylogenetic tree for tree species using the most comprehensive seed-plant phylogeny (the ALLMB tree93). This most comprehensive megatree combines a backbone tree...*".

Line 351-352: I love the idea of partitioning the angiosperm and gymnosperm phylogenies based on varying evolutionary histories

RE: Thank you.

Line 359-362: Understandably, the analyses in this paper are based on PE, but the methods also show that PD was calculated, and a good portion of the methods section was dedicated to explaining PD calculations, but PD results were omitted in the paper. I would recommend including them in the supplementary information.

RE: We did consider including the PD results initially, but after carefully consideration, we made the decision to exclude them at last. Our rationale behind this choice was to strike a balance between the amount of information presented and the clarity of the paper's logic. We felt that including PD-related result might introduce unnecessary distractions and potentially make it more challenging for readers to follow the main arguments and findings of the study. Therefore, we opted to remove all PD-related content in the revised version, with the aim of improving the overall understandability of the study. We sincerely hope that the reviewer will agree with our decision.

Line 370-371 – PD correlates strongly with species richness, but not so with PE. The findings of this study also show clear incongruence between PE and species richness patterns. And I could not find the place in the paper where this was "mentioned earlier" as used in this sentence. Please revise.

RE: Thank you for your attention to detail. As stated previously, we decided to exclude PD results in a later version, thus the phrase "mentioned earlier" was no longer applicable. In this revised version, we have rewritten the sentences to reflect the removal

of PD results.

Line 418: would the choice to extract data associated with human modification index in a different resolution (1 km) not affect the results of the study? Secondly, is this anthropogenic activity proxy layer better than human footprint index?

RE: During the analysis phase, we contemplated using the same resolution for the HMI as the other variables. However, since our intention was not to directly compare the HMI with other variables and we specifically aimed to examine the precise similarity of human pressure among the four groups, we made the decision to retain the original resolution of the HMI data. This choice allowed us to focus on assessing the relative levels of human pressure within and between the groups accurately.

In the paper presenting the HMI data, Kennedy et al. (2019) described that the HMI was developed using a larger number of recent global-scale datasets (with a median year of 2016) and a broader range of anthropogenic drivers (13 in total). This approach aimed to capture the spatial extent, intensity, and co-occurrence of human activities, many of which have significant direct or indirect impacts on biodiversity. In addition, unlike the ad hoc categorical scoring employed by the human footprint map, the HMI is a cumulative map that allows for thresholding along a continuous gradient of land modification values. This feature provides a more nuanced representation of human-induced land modifications.

For further details and a comprehensive comparison between the HMI and the human footprint map, we recommend referring to the supporting information of Kennedy et al. (2019). This source can provide a more in-depth understanding of the methodologies and differences between these two measures of human impact on the environment: <https://onlinelibrary.wiley.com/doi/full/10.1111/gcb.14549>.

Paper cited:

Kennedy CM, Oakleaf JR, Theobald DM, Baruch-Mordo S, Kiesecker J. Managing the middle: A shift in conservation priorities based on the global human modification gradient. *Glob. Chang. Biol.* 25, 811–826 (2019).

Line 441-442 – in my opinion, the half-earth proposal is unrealistic, given the human modification of the earth since prehistory. This proposal is incongruent with literature showing that 85% of the world's plant diversity is located in roughly one-third of the land surface. Moreover, this proposal is not sensitive to the impact it will have on the poor people living in developing economies (See Pimm et al. 2018, *Science*). I would have expected the CBD's 30x30 to be considered here, rather than the half-earth proposal.

RE: We agree with the reviewer that the half-earth proposal is an ambitious yet hard-to-achieve goal. However, we believe this contentious proposal offers some insights on future ambitions to ensure global protection of biodiversity, as explicitly discussed by E.O. Wilson in his "Half-Earth" book, and is relevant in policy as it was agreed at COP15 for the 2050 Vision for Biodiversity "(Lines 141-142) *he Half-Earth target56, being an*

overarching global goal for the 2050 Vision for Biodiversity agreed at COP15.”

In addition, we appreciated the reviewer's suggestion of considering the CBD's 30 x 30 target. We thus carried out further analysis to implement this. Briefly, we generated the top 30% tree diversity priority areas from our previous work (Guo et al., 2022), and checked the overlaps between the PE hotspots and this top 30% priority areas. Hence, the new version of Fig. 5 includes the CBD's 30 x 30 results. We further address the comments of Reviewer 3 about the plot style and used a grouped bar plot instead of the Doughnut plot of the older version. The relevant results were updated accordingly “(Lines 234-247) *Existing protected areas have limited protection capacity to tree phylogenetic endemism hotspots. Only 7.4% and 8.7% of angiosperm-only and joint PE hotspots are located in grid cells with existing protected areas (Fig. 5). In addition, no gymnosperm-only hotspots are protected by existing protected areas. Implementing the three conservation prioritization frameworks for tree species diversity would strongly enhance the protection level for all PE hotspots (Fig. 5). By focusing conservation efforts on the top 17% priority areas, substantial increases in protection percentages can be achieved. Specifically, the protection percentage would increase to 67.9%, 36.0%, and 78.6% PE hotspots for angiosperm-only, gymnosperm-only, and joint hotspots encompassing both angiosperm and gymnosperm, respectively. Expanding the conservation scope to include the top 30% priority areas would result in greater protection percentages: 90.4%, 68%, and 92.1% PE hotspots for angiosperm-only, gymnosperm-only, and joint hotspots, respectively. Notably, by safeguarding the top 50% priority areas, these hotspots would be almost entirely protected (≥96%; Fig. 5). Meanwhile, these priority frameworks would strongly increase the protection status of non-hotspot areas from 6.1% (existing protected areas) to 60.9% (top 50% priority areas) as well.*”

Paper cited:

Guo WY, Serra-Diaz JM, Schrod F, et al. High exposure of global tree diversity to human pressure. *Proc. Natl. Acad. Sci.* 119, e2026733119 (2022).

The drivers of floristic diversification vary for the global tropical regions. For example, the Amazon Basin enjoyed the Great American Biotic Interchange, which the remaining tropical regions did not experience. Could these unique biogeographical events have impacted the PD and PE patterns?

RE: Thanks for the question. Although there was certainly immigration into the Amazon Basin from North America via the GABI, a study by Antonelli et al. (2018) showed that more dispersal occurred the other way, that is, Amazonia is the main source for Neotropics. In addition, given the dates for the closure of the Central American Seaway range from 15-2.4 million years ago, and biotic exchanges were ongoing (and maybe even earlier) throughout this interval, this is incredibly hard to test as a "single event" (Bloch et al., 2016; Kirillova et al., 2019). Thus, the impact has likely been relatively limited.

Papers cited:

Antonelli A. et al. Amazonia is the primary source of Neotropical biodiversity. *Proc. Natl. Acad. Sci.* 115, 6034-6039 (2018).

Bloch JI et al. First North American fossil monkey and early Miocene tropical biotic interchange. *Nature* 533, 243-246 (2016).

Kirilova V, Osborne AH, Störling T and Frank M, Miocene restriction of the Pacific-North Atlantic throughflow strengthened Atlantic overturning circulation. *Nat. Commun.* 10, 4025 (2019).

Reviewer #3 (Remarks to the Author):

Review NCOMMS-23-13110-T

Guo et al. used a large dataset of global tree distribution to identify areas of phylogenetic endemism. They studied the relationship of phylogenetic endemism with present climate as well as climate change since the Late Miocene and the Last Glacial Maximum, respectively. They also looked at whether hotspots of phylogenetic endemism are affected by human land modification and protection at present and if they will experience strong climatic change under future change scenarios. They identified both areas of paleo- and neoendemism, which mostly overlapped between angiosperms and gymnosperms. Current precipitation was most strongly correlated with phylogenetic endemism among climate variables. Importantly, the authors found that areas of high phylogenetic endemism were poorly covered by existing protected areas but strongly impacted by human land use. Climate change, especially increased rainfall, will also disproportionately affect hotspots of tree phylogenetic endemism.

The study is original and significant in (i) studying a key functional group of organisms, which as the authors note in the discussion, can serve as a guide for conserving biodiversity more generally; (ii) employing a large dataset and asking relevant and clear questions regarding past, present and future of tree diversity.

The authors use appropriate statistical methods based on established metrics of phylogenetic endemism and spatial regression models.

The conclusions are valid given the analyses.

The text is well structured and easy to follow; some copy-editing is necessary and some figures could be simplified (see below).

References are appropriate although the authors may want to consider Vasconcelos et al. (2022), see below, and the original Stebbins reference discussed therein, for their "cradles" and "museums" metaphors.

RE: We appreciated the reviewer's positive and encouraging feedbacks, and thank for your considerable comments. After carefully reviewing Vasconcelos et al. (2022)'s paper, we acknowledge that the metaphors of cradles and museums, as originally described by Stebbins (1974), were not accurately reflected in our previous version. As a result, we have made necessary modification in the text accordingly, such as in Lines 74-77 "where

we first introduced the cradles and museums, we changed them to *“Specifically, areas of paleo-endemism represent potential biodiversity centers that harbour lineages which diverged or immigrated relatively deep into the past, but became extinct elsewhere while persisting within these regions. In contrast, neo-endemism characterizes biodiversity centers where recently diverged lineages are concentrated.”*

General comments

The abstract mentions very broadly "coupling not just to current climate, but also to paleo-climatic stability" as results. This could be more detailed, citing some of the specific effects found.

RE: We added the timeframe of the paleoclimate in the abstract, now it reads *“(Lines 48-49) ...with coupling not just to current climate, but also to climatic stability during millions of years back in time.”*

In the abstract and introduction, the authors seem to confound rarity and endemism. They should probably stick with "endemism" throughout for clarity, although they could mention that endemism is strongly correlated with rarity.

RE: We removed “rarity” in the abstract and stuck to “endemism”. While in the introduction, we introduced “evolutionary rarity” was represented as “phylogenetic endemism”: *“(Lines 67-70) To capture the evolutionary rarity within a given area, it has recently been proposed to quantify the degree to which phylogenetic diversity are restricted to a particular geographic area, i.e., phylogenetic endemism (PE).”*

The authors mention "cradles" and "museums" of biodiversity both in the introduction and the discussion. These metaphors introduced by Stebbins is often used, but the authors may want to consult the excellent review of Vasconcelos et al. (2022, DOI: [10.1086/717412](https://doi.org/10.1086/717412)) to get these metaphors right, if they choose to keep them.

RE: As mentioned earlier, we have now chosen to eliminate the references to cradles and museums in our revised vision. Instead, we have consistently utilized the terms “center of paleo- endemism” and “center of neo-endemism” to describe the respective centers of biodiversity. We appreciate your valuable paper suggestion, which has further guided our revisions and enhanced the clarity of our work.

The authors introduce two opposing hypotheses on the effect of climatic stability on diversification in lines 82 and 87. The text would benefit if these hypotheses (with potentially their consequences for conservation) were framed more explicitly. Again, Vasconcelos et al. 2022 could help here.

RE: We appreciate the reviewer's insightful suggestion. Taking into account the ideas presented in Vasconcelos et al. (2022), we have made modifications to the relevant sentence, with the paper cited there. Now the sentences read *“(Lines 91-102) Regions characterized by long-term climatic stability over geological time scales, coupled with optimal conditions for plant growth, have played a significant role in driving high rates of speciation and low rates of extinction. These conditions, in turn, increase the likelihood of*

immigration, resulting in higher levels of PE and the presence of either paleo-endemism or neo-endemism centers. Conversely, regions that have experienced pronounced climate oscillations, such as those occurring during glacial-interglacial cycles, are expected to exhibit both high speciation and extinction rates. This dynamic leads to substantial species turnover, and reduced chance for immigration, thereby could contribute to the formation of neo-endemism centers. Up to now, an explicit test of these hypotheses for global tree PE centers is missing, limiting our understanding of the vulnerability of tree PE hotspots, representing long-term importance for tree survival and diversification, to current human and future climate change threats^{52,53}, and from delineating efficient conservation planning.”

Maybe I am missing this, but are the calculated PE values available as dataset? This may be useful for conservation planners.

RE: Indeed, we have already uploaded the relevant data to the GitHub folder, and are planning to get this folder open and mirrored on Zenodo when the final version is approved, as shown in the next response.

Could the authors make the data and code available in a permanent repository (e.g. Zenodo), not just GitHub?

RE: Yes, we are planning to make the data and code open at both GitHub and Zenodo. We thus modified the “Data and code availability” section as “(Lines 533-535) *The R codes for the analyses are available at on Github (https://github.com/wyeco/tree_PE_conservation-threats) and are mirrored on Zenodo (link will be updated at the final stage).*”

Specific things noticed in passing

L 34: evolutionary -> evolutionarily

RE: Corrected as suggested.

Fig 1: resolution may need to be increased

RE: We increased the font size of Fig. 1 and used a higher resolution (600 dpi) in the version. Meanwhile, we will upload it in a tif file along with the revision.

Fig 3: The arrows are a bit confusing und unnecessary. Maybe best to remove them, as the violin and box plots themselves already show the relative difference.

RE: We replotted Fig. 3C and 3D using the un-transformed data and removed the arrows meanwhile, thanks.

L 183: significant -> significance

RE: Corrected as suggested.

Fig 4, small letters: font sizes really difficult to interpret. Why not just add the size of each group above or below the violins?

RE: We replotted the figure and used the same font size to improve the readability.

Fig. 5: Doughnuts really not easy to interpret, especially for the gymnosperms. A series of simple bar plots would perhaps work best here.

RE: We have taken the reviewer's advice and recreated Fig. 5 using grouped bar plots, also in response to Reviewer 2's suggestion of considering the CBD's 30 x 30 target. We fully concur with the reviewer's assessment that the revised figure strikes a balance between simplicity and information richness.

L 330 and onwards: Several regional distribution datasets were used. Is there more uncertainty around mapped distribution in areas not covered by these datasets, and could you quantify this?

RE: During the compilation of the tree species dataset, our primary aim was to gather as many occurrence records as possible using widely-used global databases. However, we encountered a situation where certain regional occurrence data, which had undergone thorough verification, had not been submitted to these global databases. To ensure comprehensive data coverage, we made the decision to include these regional datasets in order to complement the existing data.

Although we did not quantitatively assess these regional datasets, we conducted several external validations on the compiled global tree species distribution data. These validations were thoroughly described in our response to Reviewer 2, highlighting our efforts to ensure the reliability and accuracy of the dataset.

Considering the above factors, we believe that the inclusion of regional distribution datasets in our global tree species distribution data is reasonable and contributes to a more comprehensive representation of tree species occurrences worldwide.

L 361: matrices -> probably "metrics" intended here?

RE: Thanks for pointing this out. As mentioned in our response to Reviewer 2, we have made the decision to remove all PD-related content from the revised manuscript. As a result, the sentence in question has been eliminated in the latest revision.

L 405: Where was the Late Miocene data extracted from? Also from CHELSA?

RE: The Late Miocene data was obtained from the previous work of our coauthor, Dr. Matthew J. Pound, and the citation is Pound, M. J. et al. A Tortonian (Late Miocene, 11.61-7.25Ma) global vegetation reconstruction. *Palaeogeogr. Palaeoclimatol. Palaeoecol.* 300, 29–45 (2011). This is referenced in the methods section "(Lines 448-449) *Specifically, the late Miocene climate (11.61 – 7.25 Mya) was used to represent the warmer pre-glacial climate compared to the present day (hereafter Miocene)*⁴⁵".

Paper cited:

45. Pound, M. J. et al. A Tortonian (Late Miocene, 11.61-7.25Ma) global vegetation reconstruction. *Palaeogeogr. Palaeoclimatol. Palaeoecol.* **300**, 29–45 (2011).

REVIEWERS' COMMENTS

Reviewer #1 (Remarks to the Author):

The authors of the manuscript entitled "Climate change and land use threaten global hotspots of phylogenetic endemism for trees" have provided satisfactory answers to all the concerns I had with the previous version. I also feel that they have adequately addressed the concerns raised by the other two reviewers. Therefore, I feel that the manuscript has been significantly improved and I recommend that it be accepted for publication.

Reviewer #2 (Remarks to the Author):

Dear Authors,

I commend the efforts that went into improving the quality of the paper, given the feedback from reviewers. I believe this paper will make a nice contribution to biogeography and conservation disciplines.

I have one more fundamental issue which needs to be addressed. The authors mentioned that some species were manually added to the phylogeny. I think transparency is lacking in this section. First, the authors need to specify how many taxa were manually added (or should I say, grafted) into the phylogeny. Secondly, what method was used to determine the branch lengths of those manually added taxa?

I am sure the authors are aware that evolutionary biologists are divided over the idea of manually adding taxa with no DNA sequences to a phylogeny. I would not want to distract or delay this paper over such philosophical debates. But, it is absolutely necessary to explain why and how it was done in your case - for transparency and reproducibility. If it is not too much of a trouble, the authors may want to consider sensitivity analysis to assess the effect of the manually-added taxa on the overall findings of this study. I leave it to the authors to decide on doing such sensitivity analysis.

Best wishes!

Reviewer #3 (Remarks to the Author):

Congratulations on a thorough revision and nice paper. I am happy with this version overall and would recommend it for publication.

However, I still think the abstract should state more specifically the effects found (which were the important drivers? Temperature, precipitation, change since LGM or Miocene? Same for angiosperms and gymnosperms?), rather than the vague "coupling not just to current climate, but also to climatic stability across". This would help the reader.

REVIEWERS' COMMENTS

Reviewer #1 (Remarks to the Author):

The authors of the manuscript entitled "Climate change and land use threaten global hotspots of phylogenetic endemism for trees" have provided satisfactory answers to all the concerns I had with the previous version. I also feel that they have adequately addressed the concerns raised by the other two reviewers. Therefore, I feel that the manuscript has been significantly improved and I recommend that it be accepted for publication.

RE: Thank you for your kind approval of our work.

Reviewer #2 (Remarks to the Author):

Dear Authors,

I commend the efforts that went into improving the quality of the paper, given the feedback from reviewers. I believe this paper will make a nice contribution to biogeography and conservation disciplines.

RE: Thank you for your positive assessment of our revision.

I have one more fundamental issue which needs to be addressed. The authors mentioned that some species were manually added to the phylogeny. I think transparency is lacking in this section. First, the authors need to specify how many taxa were manually added (or should I say, grafted) into the phylogeny. Secondly, what method was used to determine the branch lengths of those manually added taxa?

I am sure the authors are aware that evolutionary biologists are divided over the idea of manually adding taxa with no DNA sequences to a phylogeny. I would not want to distract or delay this paper over such philosophical debates. But, it is absolutely necessary to explain why and how it was done in your case - for transparency and reproducibility. If it is not too much of a trouble, the authors may want to consider sensitivity analysis to assess the effect of the manually-added taxa on the overall findings of this study. I leave it to the authors to decide on doing such sensitivity analysis.

RE: Thanks for pointing this out. We agree with the reviewer that more information is needed for the phylogeny. There were 5,791 species (10.7% of a total of 54,020 species in our global tree phylogeny) that were manually added into the synthesis phylogeny for seed plants. We pruned the phylogeny to generate a phylogeny for 41,835 tree species used in this study. We agree with the reviewer that the generated phylogeny had some unsolved nodes due to species that were manually added into the phylogeny and unsolved nodes in the synthesis phylogeny for seed plants. However, these unsolved nodes are unlikely to bias the global analyses of tree phylogenetic patterns. Generating a phylogeny for a group of species by pruning from a synthesis tree has been widely used in ecological analyses. Recently, Li *et al.* (2019) also proved that the common community phylogenetic analyses based on a synthesis tree and a purpose-built resolved tree produce consistent

results (see Ecology 100(9): e02788; <https://doi.org/10.1002/ecy.2788>).

We have specified how many species were manually added into the phylogeny, and justified the validity of the phylogeny “(lines 380-387) *As 5,791 tree species in our 54,020 tree species dataset were missing from the megatree, they were manually added according to its genus or family, a method widely applied in similar studies. We then pruned the phylogeny to contain only species with distribution data (i.e., 41,835). Although the generated phylogeny contains some polytomies, this is unlikely to bias the global analyses of phylogenetic patterns here, as previous study had found that a phylogeny generated by pruning from a synthesis tree have consistent results in community phylogenetic analyses with those based on a purpose-built phylogeny based on gene sequence data (Li et al., 2019)*”.

Paper cited:

D. Li, et al., For common community phylogenetic analyses, go ahead and use synthesis phylogenies. Ecology 100, e02788 (2019).

Best wishes!

Reviewer #3 (Remarks to the Author):

Congratulations on a thorough revision and nice paper. I am happy with this version overall and would recommend it for publication.

However, I still think the abstract should state more specifically the effects found (which were the important drivers? Temperature, precipitation, change since LGM or Miocene? Same for angiosperms and gymnosperms?), rather than the vague "coupling not just to current climate, but also to climatic stability across". This would help the reader.

RE: We appreciated the reviewer's positive remarks on our revision. We agree with the reviewer that more detailed results should be stated in the abstract, and we further revised the abstract based on the reviewer's advice as “(line 49-52) *with current climate being the strongest driver, and climatic stability across thousands to millions of years back in time as a major co-determinant*”. Nonetheless, constrained by the abstract's word limit, we regretfully cannot delve deeper here. We hope these modifications will aid readers in swiftly comprehending the primary outcomes and rise their interests to explore the details of our research.